# REASONING-ENHANCED LARGE LANGUAGE MODELS FOR MOLECULAR PROPERTY PREDICTION

## ABSTRACT

Molecular property prediction is crucial for drug discovery and materials science, yet existing approaches suffer from limited interpretability, poor cross-task generalization, and lack of chemical reasoning capabilities. Traditional machine learning models struggle with task transferability, while specialized molecular language models provide little insight into their decision-making processes. To address these limitations, we propose **MPPReasoner**, a multimodal large language model that incorporates chemical reasoning for molecular property prediction. Our approach, built upon Qwen2.5-VL-7B-Instruct, integrates molecular images with SMILES strings to enable comprehensive molecular understanding. We develop a two-stage training strategy: supervised fine-tuning (SFT) using 16,000 high-quality reasoning trajectories generated through expert knowledge and multiple teacher models, followed by Reinforcement Learning from Principle-Guided Rewards (RLPGR). RLPGR employs verifiable, rule-based rewards that systematically evaluate chemical principle application, molecular structure analysis, and logical consistency through computational verification. Extensive experiments across 8 datasets demonstrate significant performance improvements, with MPPReasoner outperforming the best baselines by 7.91% and 4.53% on in-distribution and out-of-distribution tasks respectively. MPPReasoner exhibits exceptional cross-task generalization and generates chemically sound reasoning paths that provide valuable insights into molecular property analysis, substantially enhancing both interpretability and practical utility for chemists. Code is available at `https://anonymous.4open.science/r/MPPReasoner-12687`.

## 1 INTRODUCTION

Molecular property prediction serves as a cornerstone in modern drug discovery and materials science, enabling researchers to computationally estimate critical molecular characteristics such as bioavailability, toxicity, and therapeutic efficacy before costly experimental validation (Vamathevan et al., 2019; Schlander et al., 2021; Dara et al., 2022; Sadybekov & Katritch, 2023). Traditional experimental approaches for determining molecular properties are prohibitively expensive and time-consuming, often requiring weeks to months and costing thousands of dollars per compound (Paul et al., 2010; DiMasi et al., 2016; Wieder et al., 2020). For instance, a single ADMET (Absorption, Distribution, Metabolism, Excretion, Toxicity) screening can cost upwards of $10,000 per molecule, making it impractical to evaluate the millions of compounds in chemical space (Dong et al., 2018; Ferreira & Andricopulo, 2019). This bottleneck has driven the urgent need for accurate and efficient computational models that can predict molecular properties at scale (Shen & Nicolaou, 2019).

Despite decades of research, current molecular property prediction approaches face fundamental limitations that hinder their practical adoption. Early computational methods relied on hand-crafted molecular descriptors and traditional machine learning algorithms, which struggle with task transferability and require extensive feature engineering for each new application (Karelson et al., 1996; Rogers & Hahn, 2010; Wu et al., 2018). More recent advances have introduced specialized molecular models, including graph neural networks (GNNs) (Xie & Grossman, 2018; Zhou et al., 2023) and molecular language models (Chithrananda et al., 2020; Pei et al., 2024; Liu et al., 2024a), which have achieved impressive performance by learning molecular representations directly from graph structures or SMILES strings. Recent work has further explored interpretable molecular reasoning through large language models (Li et al., 2024b; Park et al., 2024; Kim et al., 2025; Wang et al.,

2025a), enabling models to generate natural language explanations for their predictions. However, a critical challenge remains: ensuring the **quality and verifiability** of these reasoning processes. When models generate explanations, there is no systematic way to verify whether the reasoning is chemically sound, whether predictions align with the stated logic, or whether structural analyses are accurate—limiting the model's utility in real-world drug development pipelines where trustworthy and reliable explanations are crucial for decision-making.

The fundamental limitation shared by all existing molecular property prediction methods is the absence of **effective chemical reasoning**—the ability to analyze molecular structures, identify relevant functional groups, apply chemical principles, and provide coherent explanations for property predictions. When experienced chemists evaluate molecular properties, they follow a structured reasoning process: examine the molecular structure to identify key functional groups (e.g., hydroxyl groups affecting solubility (Silverman & Holladay, 2014)), consider relevant chemical principles (e.g., Lipinski's Rule for drug-likeness (Lipinski, 2016)), analyze structure-activity relationships (McKinney et al., 2000), and synthesize these insights to make informed predictions (M. Bran et al., 2024a; Das & Rad, 2020). This reasoning capability is crucial not only for accuracy but also for trust and adoption in practical applications, as chemists need to understand the rationale behind predictions to make informed decisions about lead compound optimization and safety assessments.

To address these fundamental limitations, we propose MPPReasoner, a novel multimodal framework that introduces systematic reasoning quality verification into molecular property prediction. MPPReasoner represents the first systematic attempt to not only generate but also computationally verify domain-specific reasoning quality for molecular property prediction, enabling models to analyze molecular structures, apply established chemical principles, and provide human-interpretable explanations during the prediction process. Built upon Qwen2.5-VL-7B-Instruct (Bai et al., 2025), MPPReasoner integrates multimodal molecular representations by combining 2D molecular images with SMILES strings, enabling comprehensive structural understanding from both visual and textual modalities. Our training methodology employs a two-stage strategy to progressively develop chemical reasoning capabilities: 1) Supervised Fine-Tuning (SFT) with carefully curated reasoning trajectories generated through expert knowledge and teacher models, establishing foundational reasoning patterns; 2) Reinforcement Learning from Principle-Guided Rewards (RLPGR), a novel reward framework that leverages verifiable, rule-based feedback to enhance chemical reasoning quality. Unlike traditional reinforcement learning (RL) approaches, RLPGR decomposes chemical reasoning into hierarchical reward components that evaluate logical consistency, chemical principle application accuracy, and molecular structure analysis precision through computational verification.

Extensive experiments on 8 diverse molecular property prediction datasets demonstrate the effectiveness of our approach, achieving substantial performance improvements with average ROC-AUC scores (Hanley & McNeil, 1982; Fawcett, 2006) of 0.8190 on in-distribution (ID) tasks and 0.7977 on out-of-distribution (OOD) tasks, outperforming the best existing baselines by 13.4 and 5.5 percentage points respectively. Notably, our model exhibits exceptional OOD generalization capabilities, with particularly significant improvements on OOD datasets where many specialist models lack evaluation capability. Through expert evaluation and detailed case studies, we demonstrate that our approach produces chemically sound explanations that provide valuable insights into molecular property relationships. The main contributions of this work are as follows:

- We successfully introduce chemical reasoning capabilities into molecular property prediction tasks through MPPReasoner, representing a systematic approach to enable structured analysis, chemical principles application, and mechanistic explanations during the prediction process.

- We propose a comprehensive training strategy that combines high-quality reasoning trajectories SFT and RLPGR, a novel hierarchical reward framework targeting chemical reasoning quality through verifiable, rule-based feedback on logical consistency, comparative analysis, chemical principle usage and molecular structure analysis .

- We construct a carefully curated dataset of chemical reasoning trajectories generated through expert knowledge and few-shot prompting with multiple teacher models, providing a valuable resource for training reasoning-capable molecular property prediction models.

- We demonstrate significant performance improvements across 8 datasets with superior OOD generalization, while providing enhanced interpretability through expert-validated reasoning paths that offer insights for chemists in real-world applications.

## 2 RELATED WORK

This section reviews prior research on machine learning for molecular representations, multimodal language models in chemistry, and reasoning capabilities in LLMs, which are foundational to our proposed framework for training reasoning LLMs tailored to molecular property prediction.

**Machine Learning for Molecular Representation.** GNNs have evolved as a dominant paradigm for molecular graph representation, progressing from early convolutional (Kearnes et al., 2016; Schütt et al., 2018) and message-passing (Gilmer et al., 2017) frameworks to sophisticated 3D-aware models (Gilmer et al., 2017; Li et al., 2022; Liu et al., 2023c; Zhou et al., 2023), enabling robust applications in property prediction, virtual screening, and drug discovery (Alves et al., 2022). In parallel, specialized molecular language models have reframed molecular structures as textual sequences such as SMILES strings(Weininger, 1988), with models like MolecularGPT (Liu et al., 2024a) and BioT5-Plus (Pei et al., 2024) supporting few-shot adaptation and multi-task learning for diverse chemical and biological tasks (Li et al., 2024a; Liu et al., 2024b; 2023b).

**Language Models for Chemistry.** The emerging trend of multimodal LLMs in chemistry integrates diverse data types—such as SMILES strings and molecular graphs to address unimodal limitations, as seen in foundational molecular-text models (Lee et al., 2025; Li et al., 2025b; Liu et al., 2023b), instruction-tuned assistants (Cao et al., 2025), and tool-augmented systems (M. Bran et al., 2024b), enhancing robustness in property prediction (Edwards et al., 2022), molecular design (Lee et al., 2025), and synthesis planning (Shi et al., 2023; Liu et al., 2024b). Recent works (Li et al., 2024b; Park et al., 2024; Kim et al., 2025; Wang et al., 2025a; Zheng et al., 2025) have enabled interpretable molecular reasoning through chain-of-thought. Our work complements these advances by introducing computational verification to ensure reasoning quality and chemical correctness.

**Reasoning in Large Language Models.** Reasoning capabilities have demonstrated remarkable efficacy in commercial LLMs, particularly through chain-of-thought as exemplified in OpenAI's o1 series and other advanced models (Wei et al., 2022; Jaech et al., 2024; Team et al., 2023; Priyanshu et al., 2024). Training these abilities leverages RL techniques, like Proximal Policy Optimization (Schulman et al., 2017) in RL from Human Feedback (RLHF) (Ouyang et al., 2022) for preference alignment, and efficient extensions like Group Relative Policy Optimization (GRPO) (Shao et al., 2024) with outcome-based rewards and Reinforcement Learning with Verifiable Rewards (RLVR), improving generalization on complex tasks (Wang et al., 2025c). These RL advancements motivate our adaptation for chemical-specific reasoning in the field of molecular property prediction.

## 3 METHODOLOGY

MPPReasoner cultivates chemical reasoning capabilities in multimodal large language models through a structured approach illustrated in Figure 1. We first construct high-quality reasoning trajectories that demonstrate expert-level chemical analysis patterns. These trajectories are then used in a two-stage training framework: SFT establishes foundational reasoning abilities, followed by RL guided by our novel Principle-Guided Reward mechanism.

### 3.1 MULTIMODAL MOLECULAR PROMPT DESIGN

To provide comprehensive molecular understanding, we employ a multimodal input representation that integrates 2D molecular images with their corresponding SMILES strings. This dual representation enables the model to capture both the sequential chemical information encoded in SMILES and the spatial structural relationships depicted in molecular visualizations.

As shown in Appendix A, our prompt engineering strategy comprises four essential components: **[Role Definition]** instructs the model to act as an expert chemist specializing in molecular property prediction; **[Task Description]** provides task-specific instructions outlining the prediction objective and requirement for step-by-step reasoning; **[Few-Shot Examples]** are dynamically retrieved by identifying the top-5 most similar molecules from the training set using Tanimoto similarity (Bajusz et al., 2015) based on Morgan fingerprints (Cereto-Massagué et al., 2015); and **[Multimodal**

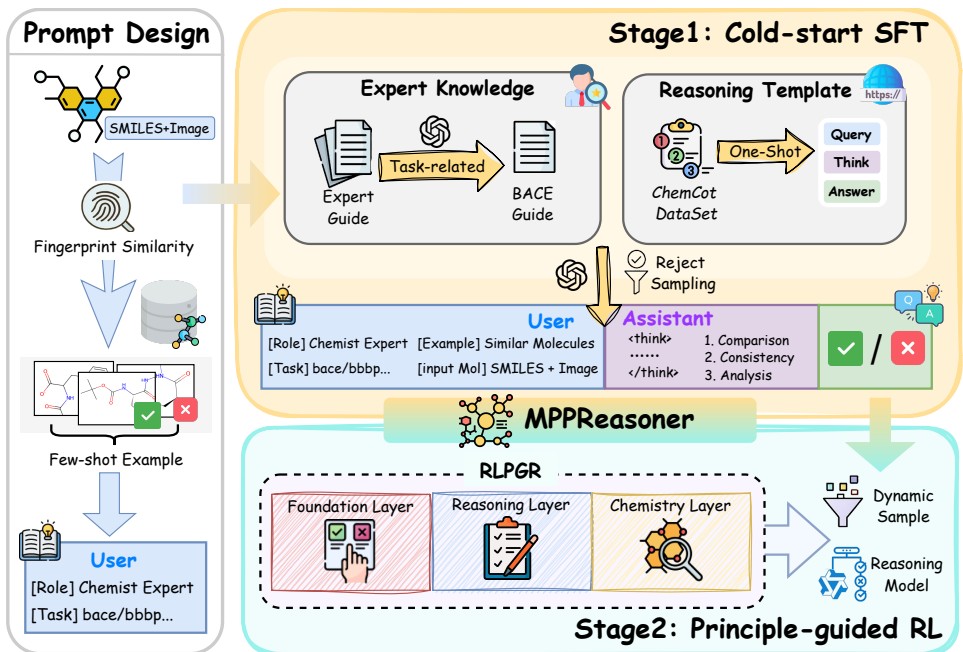

Figure 1: Overview of MPPReasoner framework.

**Molecule]** includes both the rendered 2D molecular structure image and the corresponding SMILES string, providing complementary perspectives on molecular characteristics.

## 3.2 TWO-STAGE TRAINING FRAMEWORK FOR CHEMICAL REASONING

### 3.2.1 STAGE 1: REASONING TRAJECTORY CONSTRUCTION FOR COLD START SFT

The first stage establishes foundational chemical reasoning capabilities through high-quality data construction and supervised learning. We begin by constructing a comprehensive dataset of chemical reasoning trajectories through two complementary approaches:

**Multi-Source Reasoning Data Construction.** We leverage powerful large general-domain reasoning models as teacher models to generate high-quality chemical reasoning patterns through two complementary approaches:

- *ChemCoT-Based One-Shot Generation:* We utilize exemplars from the ChemCoT dataset (Li et al., 2025a) as one-shot demonstrations, instructing teacher models to emulate the step-by-step analytical style demonstrated in chemical chain-of-thought examples.

- *Expert-Guided Task-Specific Generation:* We invited 5 domain experts from different chemical disciplines to independently draft reasoning guides covering fundamental principles for various molecular properties. These guides underwent rigorous cross-validation, ensuring high quality. The consolidated expert knowledge is then processed by GPT-4o (OpenAI et al., 2024) to extract task-specific knowledge relevant to each dataset (e.g., BACE, BBBP (Wu et al., 2018)). The extracted principles serve as domain-specific prompts in generating reasoning trajectories.

**Quality Control and Data Curation.** We employ rejection sampling to ensure trajectory quality, accepting only those instances where teacher models produce correct predictions (True/False). This process yields 16,000 high-quality reasoning trajectories for SFT.

Using these curated reasoning trajectories, we perform SFT on Qwen2.5-VL-7B-Instruct with standard next-token prediction loss. The reasoning generation process utilizes three state-of-the-art language models (GPT4o (OpenAI et al., 2024), DeepSeek-v3.1 (Guo et al., 2025) and Qwen2.5VL (Bai et al., 2025)) working in parallel to ensure diversity in reasoning patterns while

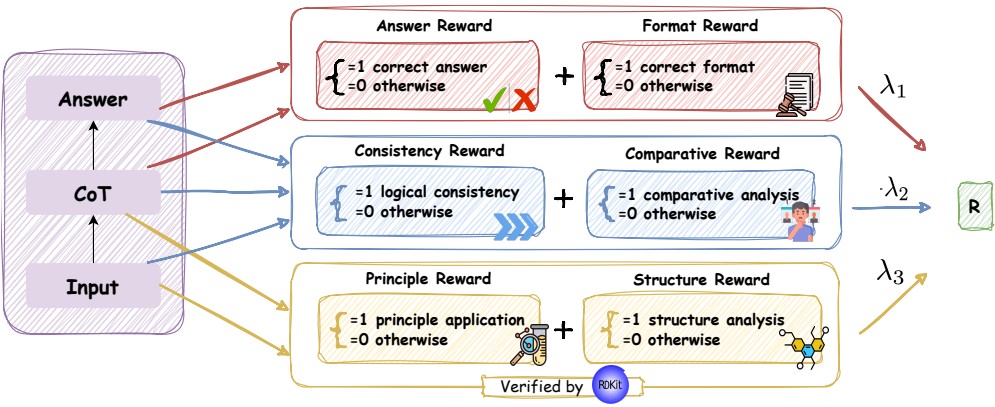

Figure 2: Illustration of RLPGR in MPPReasoner.

maintaining high quality through model complementarity. When multiple teacher models generate correct reasoning for the same instance, we randomly select one trajectory to maintain dataset diversity. This stage focuses on instruction alignment, teaching the model to follow the required format of providing step-by-step reasoning before making final predictions, while simultaneously instilling domain-specific knowledge and analytical patterns demonstrated in the expert-curated trajectories.

### 3.2.2 STAGE 2: ADVANCED REASONING REFINEMENT WITH RLPGR

While SFT establishes basic reasoning patterns, the second stage employs RLPGR to elevate the model's capabilities from imitation to exploration and refinement. Unlike traditional RLHF (Ouyang et al., 2022) approaches that rely on human preference data, RLPGR leverages verifiable, rule-based rewards (Wang et al., 2025c) derived from chemical principles and computational tools (Landrum, 2013), ensuring both scalability and domain accuracy.

Our RLPGR framework decomposes the complex cognitive process of chemical reasoning into measurable reward components across three hierarchical layers. As illustrated in Figure 2, given a molecular description $x$, reasoning trace $z$, and prediction $y$, the total reward is computed as:

$$R_{\text{total}}(x, z, y) = \lambda_1 \underbrace{(r_{\text{ans}} + r_{\text{fmt}})}_{R_{\text{foundation}}} + \lambda_2 \underbrace{(r_{\text{cons}} + r_{\text{comp}})}_{R_{\text{reasoning}}} + \lambda_3 \underbrace{(r_{\text{prin}} + r_{\text{struct}})}_{R_{\text{chemistry}}} \tag{1}$$

where $\lambda_i$ are hyperparameters controlling the relative importance of each reward component.

**Foundation Layer:** This layer ensures basic task requirements through two components:

- Answer reward $r_{\text{ans}}$ provides binary feedback based on prediction correctness.

- Format compliance reward $r_{\text{fmt}}$ verifies that outputs follow the required structure with reasoning enclosed in `<think>` tags and predictions in `<answer>` tags.

**Reasoning Layer:** This layer evaluates general reasoning quality through two key aspects:

- Logical consistency reward $r_{\text{cons}}$ measures alignment between the reasoning conclusion and final prediction through a two-step verification process: (1) detecting conclusion keywords that indicate explicit final judgment, and (2) verifying sentiment consistency between reasoning and prediction using predefined affirmative/negative term sets for True/False predictions respectively. Awards $r_{\text{cons}} = 1$ if both conditions are satisfied, otherwise $r_{\text{cons}} = 0$.

- Comparative analysis reward $r_{\text{comp}}$ encourages analogical thinking by detecting explicit mentions of few-shot example SMILES strings within the reasoning text. Awards $r_{\text{comp}} = 1$ if at least one example is referenced, otherwise $r_{\text{comp}} = 0$.

**Chemistry Layer:** This layer targets domain-specific expertise through computational verification of chemical knowledge and structural analysis accuracy. We leverage RDKit (Landrum, 2013) for molecular property computation and substructure detection to provide objective feedback on chemical reasoning quality.

- Chemical principle application reward $r_{\text{prin}}$ evaluates whether mentioned chemical concepts align with computationally derived molecular properties. For instance, when the reasoning discusses hydrophobicity, we verify this against the computed LogP value. This reward ensures that chemical principles are applied appropriately rather than superficially mentioned.

- Molecular structure analysis reward $r_{\text{struct}}$ measures the coverage of structural feature identification as $r_{\text{struct}} = |S_{\text{actual}} \cap S_{\text{pred}}|/(|S_{\text{actual}}| + \epsilon)$, where $S_{\text{actual}}$ represents the set of distinct structural feature types identified via RDKit (functional groups, ring systems, stereochemical features), $S_{\text{pred}}$ denotes the set of structural feature types mentioned in the reasoning trace $z$, and $\epsilon = 10^{-5}$ ensures numerical stability.

**Training Process.** We employ GRPO (Shao et al., 2024) for policy optimization, which maximizes the expected reward across reasoning trajectories:

$$\mathcal{L}_{\text{RLPGR}} = \mathbb{E}_{(x,z,y) \sim \pi_\theta}[R_{\text{total}}(x, z, y)] \tag{2}$$

where $\pi_\theta$ represents the policy parameterized by $\theta$. Dynamic sampling during training focuses computational resources on tractable reasoning examples, ensuring efficient convergence toward models capable of generating chemically accurate and interpretable reasoning paths. This RLPGR approach transforms the model from pattern-matching to genuine chemical reasoning through systematic principle-guided RL.

# 4 EXPERIMENTS

To comprehensively evaluate our approach, we investigate the following Research Questions (RQs):

**RQ1:** Does MPPReasoner achieve superior performance compared to existing molecular property prediction methods on both ID and OOD datasets?

**RQ2:** What are the individual contributions of the two-stage training strategy and the RLPGR reward components to the overall performance and reasoning quality?

**RQ3:** Does incorporating 2D molecular images provide meaningful advantages over text-only SMILES representations for molecular property prediction?

**RQ4:** Can our model generate high-quality reasoning paths that provide chemically meaningful insights comparable to expert-level analysis?

## 4.1 EXPERIMENTAL SETUP

**Datasets.** We evaluate MPPReasoner on 8 diverse molecular property prediction datasets to assess both ID and OOD performance. The datasets are categorized as follows:

- *ID Datasets:* We utilize four benchmark datasets from MoleculeNet (Wu et al., 2018), which is widely used to predict whether the given molecule has specific properties: BACE (1,513), BBBP (2,039), SIDER (1,427), HIV (41,127).

- *OOD Datasets:* We employ four datasets from the Therapeutic Data Commons (TDC) (Huang et al., 2021; 2022) to evaluate cross-task generalization capabilities: Bioavailability (128), CYP2C9_V (2,418), CYP2D6_V (2,626), AMES (1,456).

The ID/OOD categorization is based on whether the training set includes samples from the corresponding dataset. For training, we randomly sample 4,000 instances from the ID datasets to ensure balanced representation across different molecular properties. Test sets follow standard benchmarking protocols established in prior literature to maintain fair comparison with baseline methods.

Table 1: Performance comparison of task-specific specialist models and LLM-based generalist models on ID and OOD benchmarks. Best performance is in **bold**. Second best is underlined.

| Model | ID Performance | | | | OOD Performance | | | | Average | |
|---|---|---|---|---|---|---|---|---|---|---|
| | BACE | BBBP | SIDER | HIV | Bioavail. | C2C9_V | C2D6_V | AMES | ID | OOD |
| *# Task-specific specialist models* | | | | | | | | | | |
| Graphormer-p | 0.8575 | 0.7163 | – | 0.7788 | – | – | – | – | 0.7842 | – |
| Uni-Mol | 0.8570 | 0.7290 | 0.6590 | **0.8080** | – | – | – | – | 0.7633 | – |
| GIMLET | 0.6957 | 0.5939 | – | 0.6624 | – | – | – | – | 0.6507 | – |
| MolecularGPT | 0.7331 | 0.6822 | – | 0.6382 | – | – | – | – | 0.6845 | – |
| Mol-LLM | 0.8080 | **0.8430** | 0.7610 | 0.7650 | – | – | – | – | 0.7943 | – |
| InstructMol-GS | 0.8210 | 0.7240 | – | 0.6890 | – | – | – | – | 0.7447 | – |
| BioT5-Plus | 0.8620 | 0.7650 | 0.5201 | 0.7630 | 0.5243 | 0.4971 | 0.5321 | 0.4466 | 0.7275 | 0.5000 |
| MolXPT | 0.8840 | 0.8000 | 0.7170 | 0.7810 | 0.4749 | 0.5904 | 0.5291 | 0.6073 | 0.7955 | 0.5504 |
| 3D-MoLM | 0.7287 | 0.5141 | 0.5073 | 0.6603 | 0.6066 | 0.7081 | 0.7029 | 0.7190 | 0.6026 | 0.6842 |
| Mol-LLaMA | 0.8349 | 0.6263 | 0.5576 | 0.7249 | 0.6020 | 0.7556 | 0.7789 | 0.7928 | 0.6859 | 0.7323 |
| TxGemma | 0.6380 | 0.7102 | 0.5619 | 0.5235 | 0.5813 | 0.8075 | 0.7595 | 0.7907 | 0.6084 | 0.7348 |
| *# LLM-based generalist models* | | | | | | | | | | |
| o3-mini | 0.7891 | 0.5972 | 0.5626 | 0.6039 | 0.6246 | 0.7729 | 0.7643 | 0.8361 | 0.6382 | 0.7495 |
| DeepSeek-V3.1-Think | 0.7017 | 0.6048 | 0.5637 | 0.5938 | 0.6572 | 0.7633 | 0.7484 | 0.8218 | 0.6160 | 0.7477 |
| GPT-4o | 0.6070 | 0.6731 | 0.6347 | 0.5698 | 0.5826 | 0.5508 | 0.5902 | 0.6141 | 0.6212 | 0.5844 |
| Gemma-3-12b-it | 0.8526 | 0.6802 | 0.5652 | 0.6886 | 0.6194 | 0.7886 | 0.7780 | 0.8130 | 0.6967 | 0.7498 |
| Qwen3-8B | 0.7924 | 0.6961 | 0.6442 | 0.5706 | 0.6181 | 0.7496 | 0.7165 | 0.8392 | 0.6758 | 0.7309 |
| Qwen3-VL-8B-Instruct | 0.8757 | 0.7096 | 0.5905 | 0.7143 | 0.5919 | 0.8012 | 0.7784 | 0.8514 | 0.7225 | 0.7557 |
| Qwen3-VL-8B-Thinking | 0.8597 | 0.7055 | 0.5973 | 0.5890 | 0.6073 | 0.7883 | 0.7572 | 0.8718 | 0.6879 | 0.7562 |
| Qwen2.5-VL-72B-Instruct | 0.7764 | 0.5791 | 0.5880 | 0.7325 | 0.6388 | 0.7624 | 0.7222 | 0.8156 | 0.6690 | 0.7348 |
| Qwen2.5-VL-7B-Instruct | 0.6910 | 0.6175 | 0.5823 | 0.5125 | 0.5232 | 0.7333 | 0.6999 | 0.7667 | 0.6008 | 0.6808 |
| **MPPReasoner (Ours)** | **0.9090** | 0.7436 | **0.8280** | 0.7932 | **0.6728** | **0.8480** | **0.7950** | **0.8750** | **0.8190** | **0.7977** |

**Baseline.** We compare MPPReasoner against two categories of approaches:

- *Task-specific Specialist Models:* These models are designed for molecular property prediction: Graphormer-p (Ying et al., 2021), Uni-Mol (Zhou et al., 2023), GIMLET (Zhao et al., 2023), MolecularGPT (Liu et al., 2024a), Mol-LLM (Lee et al., 2025), InstructMol-GS (Cao et al., 2025), BioT5-Plus (Pei et al., 2024), MolXPT (Liu et al., 2023a), 3D-MoLM (Li et al., 2024b), Mol-LLaMA (Kim et al., 2025), TxGemma (Wang et al., 2025a).

- *LLM-based Generalist Models:* These include reasoning models: o3-mini (OpenAI, 2025), DeepSeek-V3.1 (Guo et al., 2025), large-scale models: GPT-4o (OpenAI et al., 2024), Qwen2.5-VL-72B-Instruct (Bai et al., 2025), Gemma-3-12b-it (Kamath et al., 2025), Qwen3-8B (Yang et al., 2025), Qwen3-VL-8B-Instruct/Thinking (Yang et al., 2025), and baseline models: Qwen2.5-VL-7B-Instruct (Bai et al., 2025) applied to molecular property prediction. All LLM-based models are evaluated under the few-shot setting described in Section 3.1

Implementation details and hyper-parameters settings are provided in Appendix C.

## 4.2 MAIN RESULTS (RQ1)

Table 1 presents the comprehensive performance comparison of MPPReasoner against state-of-the-art baselines across all 8 datasets. On ID datasets, MPPReasoner demonstrates competitive performance with specialized models while maintaining the advantage of using a smaller 7B parameter architecture. MPPReasoner achieves the best performance on challenging tasks like BACE and SIDER, indicating successful capture of complex molecular property relationships such as enzyme inhibition and side effect prediction. While some specialized models like Mol-LLM excel on specific tasks such as BBBP, these models achieve high ID performance at the expense of cross-task adaptability, completely lacking OOD evaluation capability. This specialization-generalization trade-off limits their practical utility in real-world scenarios requiring diverse molecular property assessment.

The most significant advantage emerges in OOD scenarios, where MPPReasoner substantially outperforms all baseline categories by 5.5% over the best baseline. This consistent superiority across diverse molecular property types highlights how domain-specific chemical reasoning outperforms both general reasoning capabilities and raw parameter scaling approaches. Notably, MPPReasoner achieves 0.8190 on ID tasks, significantly outperforming the best baseline by 13.4%, with partic-

Table 2: Ablation study on training stages and RLPGR reward.

| Setting | ID Performance | | | | OOD Performance | | | | Average | |
|---|---|---|---|---|---|---|---|---|---|---|
| | BACE | BBBP | SIDER | HIV | Bioavail. | C2C9_V | C2D6_V | AMES | ID | OOD |
| Base (Qwen2.5-VL-7b-Instruct) | 0.6910 | 0.6175 | 0.5823 | 0.5125 | 0.5232 | 0.7333 | 0.6999 | 0.7667 | 0.6008 | 0.6808 |
| SFT Only | 0.8558 | 0.6824 | 0.6752 | 0.7186 | 0.6625 | 0.7799 | 0.7348 | 0.8415 | 0.7330 | 0.7547 |
| RL Only (RLPGR) | 0.8142 | 0.5733 | 0.7428 | 0.5552 | 0.6632 | 0.7491 | 0.6732 | 0.7300 | 0.6714 | 0.7039 |
| SFT + $R_{\text{foundation}}$ | 0.8836 | 0.6794 | 0.8089 | 0.7556 | 0.6358 | 0.8364 | 0.7862 | 0.8536 | 0.7819 | 0.7780 |
| SFT + $R_{\text{foundation}}$ + $R_{\text{reasoning}}$ | 0.8877 | 0.7104 | 0.7981 | 0.7560 | **0.6771** | 0.8140 | 0.7795 | 0.8388 | 0.7881 | 0.7774 |
| **MPPReasoner (Ours)** | **0.9090** | **0.7436** | **0.8280** | **0.7932** | 0.6728 | **0.8480** | **0.7950** | **0.8750** | **0.8190** | **0.7977** |

ularly strong gains on challenging tasks like SIDER. The performance gap becomes even more pronounced when considering that MPPReasoner operates with significantly fewer parameters than competing large-scale models.

The results reveal fundamental differences between model categories and their limitations. Task-specific specialist models excel in familiar scenarios but completely lack cross-task generalization capabilities, while generalist models show consistent cross-task performance but suffer from insufficient domain expertise. MPPReasoner uniquely bridges this gap by embedding domain-specific reasoning rather than relying on general reasoning patterns or parameter scaling alone. The transformative impact becomes evident when comparing MPPReasoner to its base model, showing dramatic improvements of 36.36% on ID tasks and 17.17% on OOD tasks. This demonstrates that structured chemical reasoning fundamentally enhances molecular understanding beyond conventional approaches, enabling both specialist-level accuracy and generalist-level adaptability through systematic integration of chemical principles.

### 4.3 TRAINING STRATEGY ABLATION STUDIES (RQ2)

To understand the individual contributions of our two-stage training strategy and the hierarchical reward components in RLPGR, we conduct comprehensive ablation studies. Table 2 presents the systematic analysis of each component's impact on both ID and OOD performance. The results demonstrate that both SFT and RL stages contribute significantly to overall performance, with distinct advantages for different aspects. SFT alone provides substantial improvements over the base model, achieving 22.01% and 10.85% gains on ID and OOD tasks respectively, indicating that SFT with high-quality reasoning trajectories successfully instills foundational chemical reasoning capabilities. RL alone also shows meaningful improvements of 11.74% and 3.39%, demonstrating that principle-guided rewards can enhance reasoning quality independently. However, the combination of SFT + RL yields the strongest performance with 36.36% and 17.17% improvements, revealing important synergistic effects between two training stages that exceed their individual contributions.

The progressive addition of RLPGR reward components shows clear incremental benefits, validating our hierarchical design. Foundation rewards provide substantial improvements of 6.67% on ID and 3.09% on OOD over SFT alone, establishing enhanced task completion capabilities beyond basic instruction following. The Reasoning layer contributes additional 0.84% ID gains while maintaining similar OOD performance, indicating that logical consistency and comparative analysis refine reasoning quality without compromising generalization. Most importantly, the Chemistry layer delivers the largest incremental improvements of 3.92% on ID and 2.61% on OOD, confirming that domain-specific chemical principle verification is crucial for molecular property prediction tasks.

### 4.4 MULTIMODAL ABLATION STUDY (RQ3)

While SMILES encode complete molecular structure sequentially, recent work demonstrates that visual representations can capture spatial patterns and topological relationships that complement textual encoding (Zeng et al., 2022; Yin et al., 2025; Xie et al., 2025). To investigate whether 2D molecular images provide meaningful advantages beyond SMILES strings alone, we conduct systematic ablation experiments comparing text-only and multimodal MPPReasoner.

We train two variants of our framework: (1) MPPReasoner (Text): using only SMILES strings with Qwen2.5-7B-Instruct as the base model, and (2) MPPReasoner (MLLM): using both SMILES and

Table 3: Ablation study comparing text-only and multimodal versions of MPPReasoner.

| Method | ID Performance | | | | OOD Performance | | | | Average | |
|---|---|---|---|---|---|---|---|---|---|---|
| | BACE | BBBP | SIDER | HIV | Bioavail. | C2C9_V | C2D6_V | AMES | ID | OOD |
| Qwen2.5-7B-Instruct | 0.6950 | 0.5391 | 0.5820 | 0.4957 | 0.5094 | 0.7280 | 0.7132 | 0.7149 | 0.5780 | 0.6664 |
| MPPReasoner (Text) | 0.7616 | 0.6187 | 0.5599 | 0.5837 | 0.6173 | 0.6810 | 0.6771 | 0.7646 | 0.6810 | 0.7100 |
| | ↑9.6% | ↑14.8% | ↓3.8% | ↑17.8% | ↑21.2% | ↓6.5% | ↓5.1% | ↑7.0% | ↑17.8% | ↑6.5% |
| Qwen2.5-VL-7B-Instruct | 0.6910 | 0.6175 | 0.5823 | 0.5125 | 0.5232 | 0.7333 | 0.6999 | 0.7667 | 0.6008 | 0.6808 |
| MPPReasoner (MLLM) | **0.9090** | **0.7436** | **0.8280** | **0.7932** | **0.6728** | **0.8480** | **0.7950** | **0.8750** | **0.8190** | **0.7977** |
| | ↑31.5% | ↑20.4% | ↑42.2% | ↑54.8% | ↑28.6% | ↑15.6% | ↑13.6% | ↑14.1% | ↑36.3% | ↑17.2% |

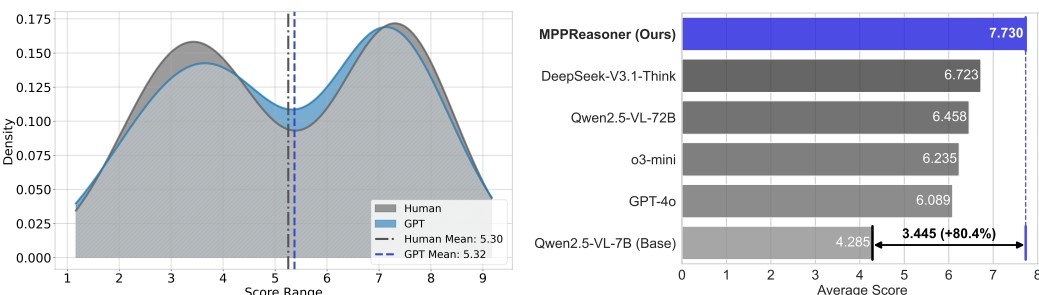

(a) Human-AI evaluation consistency        (b) Model reasoning quality scores

Figure 3: Reasoning quality evaluation results. (a) Strong consistency between automated and human assessments with $\rho = 0.82$. (b) MPPReasoner achieves the highest reasoning quality score.

2D molecular images with Qwen2.5-VL-7B-Instruct. Both variants undergo identical two-stage training (SFT + RLPGR) with the same hyperparameters and training data.

Table 3 presents the comprehensive comparison. Both versions demonstrate substantial improvements over their respective base models, confirming the effectiveness of our chemical reasoning framework regardless of modality. Notably, the multimodal version significantly outperforms the text-only variant across nearly all datasets, achieving +18.9% improvement on ID tasks and +10.5% on OOD tasks. Interestingly, while the text-only version achieves competitive performance compared to specialized baselines like BioT5-Plus and MolecularGPT, the multimodal approach represents a substantial enhancement that leverages additional visual information for improved chemical reasoning. These results validate our architectural choice to build upon multimodal LLMs, demonstrating that 2D molecular images provide meaningful complementary information beyond SMILES.

## 4.5 REASONING QUALITY EVALUATION (RQ4)

To assess whether MPPReasoner generates high-quality reasoning paths that provide chemically meaningful insights, we conduct systematic evaluation using automated assessment validated against human expert judgment. We employ a LLM-as-a-Judge (Zheng et al., 2023) framework using GPT-4o to evaluate three dimensions (Fan et al., 2025): **logical soundness, accuracy & insight, and conciseness**, each scored on a 0-10 scale with detailed rubrics. To establish reliability, we validate GPT-4o scores against human expert assessments on 60 reasoning samples from three baseline models. Figure 3(a) shows remarkable consistency between automated and human evaluations, with similar distributions and central tendencies with Spearman correlation coefficient reaches $\rho = 0.82$.

Figure 3(b) presents the comparative reasoning quality assessment across different model categories, showing average scores across the three evaluation dimensions. MPPReasoner achieves the highest score of 7.730, substantially outperforming advanced reasoning models including DeepSeek-V3.1-Think at 6.723 and o3-mini at 6.235, as well as large-scale models like Qwen2.5-VL-72B at 6.458 and GPT-4o at 6.089. Despite using a smaller 7B architecture, MPPReasoner demonstrates 15.0% improvement over the best baseline, highlighting how domain-specific chemical reasoning surpasses both general reasoning capabilities and parameter scaling approaches. The superior reasoning quality translates to practical benefits: MPPReasoner consistently identifies relevant functional groups, applies appropriate chemical principles, and provides mechanistic explanations that enable chemists

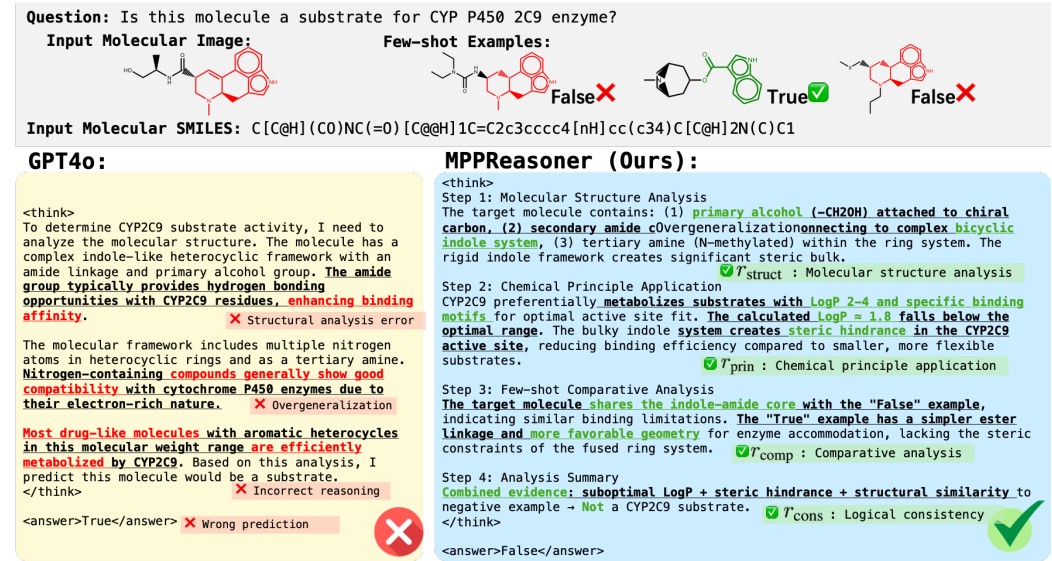

Figure 4: Case study comparison between GPT-4o and MPPReasoner for CY-P450-2C9 substrate prediction. The figure demonstrates how RLPGR training enables systematic chemical reasoning with accurate predictions, while GPT4o suffer from several errors.

to understand both what properties a molecule has and why these properties emerge from specific structural features. Detailed dimensional scores are provided in Appendix B.

# 5 CASE STUDY

To illustrate the practical benefits of our chemical reasoning approach, Figure 4 presents a representative case study comparing MPPReasoner with GPT-4o on CY-P450-2C9 substrate prediction. The comparison reveals fundamental differences in analytical quality: GPT-4o exhibits critical reasoning flaws including structural analysis errors (incorrectly assuming amide groups enhance binding affinity), overgeneralization (broadly claiming nitrogen-containing compounds show P450 compatibility), and incorrect reasoning patterns (unsupported statistical generalizations), ultimately leading to a wrong prediction. In contrast, MPPReasoner demonstrates systematic chemical reasoning through accurate molecular structure analysis (precise functional group identification), correct chemical principle application (referencing CYP2C9-specific LogP requirements and calculating steric hindrance), meaningful comparative analysis (connecting structural similarities to substrate labels), and logical consistency (integrating multiple evidence sources). This exemplifies how RLPGR's hierarchical rewards successfully cultivate domain-specific reasoning capabilities, enabling chemists to trust the model's mechanistic insights for practical applications.

# 6 CONCLUSION

This work introduces MPPReasoner, a novel multimodal large language model that systematically incorporates chemical reasoning for molecular property prediction. Through a novel two-stage training strategy combining supervised fine-tuning with RLPGR, our approach successfully bridges the gap between specialist accuracy and generalist adaptability. Extensive experiments across 8 datasets demonstrate substantial performance improvements of 20.60% and 9.93% on ID and OOD tasks respectively, with exceptional cross-task generalization capabilities where many specialist models lack evaluation capability. Beyond performance gains, MPPReasoner generates chemically sound reasoning paths that enable chemists to understand not just prediction outcomes but the underlying chemical rationale. This represents a crucial advancement toward interpretable AI systems that provide mechanistic insights grounded in established chemical principles, supporting informed decision-making in drug discovery and offering a potential blueprint for other scientific domains.

ETHICS STATEMENT

In developing MPPReasoner, we prioritized ethical considerations to ensure responsible use of our models and methodologies. Our research does not involve human subjects, and all molecular data are obtained from publicly available, copyright-compliant datasets (MoleculeNet and TDC) with appropriate research licensing. We employed rigorous quality filtering processes during data curation to minimize biased or misleading chemical information. We acknowledge that biases inherent in molecular property datasets, including overrepresentation of certain chemical scaffolds, may propagate into model outputs. While MPPReasoner is designed for beneficial applications in drug discovery and materials science, we encourage responsible use within established scientific and regulatory frameworks. Our work is conducted with commitment to research integrity, ensuring contributions remain beneficial to the scientific community while addressing ethical responsibilities of developing AI technologies for chemical applications.

REPRODUCIBILITY STATEMENT

We have made comprehensive efforts to ensure the reproducibility of MPPReasoner and our experimental findings. Our two-stage training methodology is detailed in Section 3, including the SFT process and the novel RLPGR framework with specific reward components. Complete implementation details, hyperparameter configurations, and training procedures are provided in Appendix C. The experimental setup, including dataset descriptions, baseline model configurations, and evaluation protocols, is thoroughly documented in Section 4.1 and Appendix C. All datasets used in our experiments are publicly available: the ID datasets are from MoleculeNet, and the OOD datasets are from TDC. The reasoning trajectory construction process using expert knowledge and multiple teacher models is described in Section 3.2.1, with specific prompting strategies detailed in Appendix A. Our reasoning quality evaluation methodology, including the LLM-as-a-Judge framework and human expert validation procedures, is documented in Section 4.5. We provide the complete source code for model training, evaluation, and reasoning quality assessment at `https://anonymous.4open.science/r/MPPReasoner-12687`.

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

## A  PROMPT TEMPLATES

---

**Example 1: Prompt Example for BACE Task**

**[Role]**
You are a top AI assistant specializing in molecular chemistry and drug discovery, proficient in molecular property prediction.

**[Task]**
BACE1 is an aspartic-acid protease important in the pathogenesis of Alzheimer's disease, and in the formation of myelin sheaths...
output "True" or "False".

**[Formatting]**
Place the thought process within <think></think> and then conclude your answer with <answer>True/False</answer>.

**[Example]**
<think>xxxx</think>
<answer>True/False</answer>

**[Few-shot]**

| | |
|---|---|
| ClC1=CC(=CC(Cl)=C1NC(=O)C)CNC(=[NH2+1])NC(=O)CN2C3=CC(OC)=CC=C3C=C2 | False |
| ClC1=CC(=CC(Cl)=C1NC(=O)C)CNC(=[NH2+1])NC(=O)CN2C3=CC(CC)=CC=C3C=C2 | True |
| ClC1=CC(=CC(Cl)=C1NC(=O)C)CNC(=[NH2+1])NC(=O)CN2C3=CC(F)=CC3C=C2 | False |

**[Molecule]**
ClC1=CC(=CC(Cl)=C1NC(=O)C)CNC(=[NH2+1])NC(=O)CN2C3=C(C=CC=C3)C=C2

---

**Example 2: Prompt for ChemCoT-Based One-Shot Generattion**

**Example Prompt:**

< PORMPT retrieved from OpenMol/ChemCoTDataset >

**Example Response:**

< RESPONSE retrieved from OpenMol/ChemCoTDataset >

**Prompt:**

< PORMPT likes Example 1 >

**Response:**

---

**Example 3: Prompt for Expert-Guided Task-Specific Generation**

```
< PORMPT likes Example 1 >

[Expert]

< EXPERT KNOWLEDGE refined by GPT4o >
```

**Example 4: Prompt for Logical Soundness Scoring**

```
You are a professional reasoning-evaluation expert.  Your
task is to assess the logical soundness of a large language
model's chain-of-thought when answering a question, and
assign an integer score from 0 to 10.  Focus strictly on the
logical connections between reasoning steps, not on whether
the final answer is correct.

Input:
- [Question]:  The original question.
- [Model Response]:  The model's full response, including its
chain of thought.

Scoring Dimension (Logical Soundness):
- Do reasoning steps build progressively and refer back to
earlier points?
- Is each step a reasonable extension of the previous
inference?
- Is the language coherent, with no contradictions or
confusing wording?

Scoring Scale (0-10):
- 10:  Perfect logical structure; steps are crystal-clear and
fully justified.
- 8-9:  Overall logic sound; only minor or negligible
leaps/wording issues.
- 6-7:  Main logic correct, but some jumps, insufficient
explanation, or minor conflicts.
- 4-5:  Noticeable breaks or missing key inferences, yet some
coherent logic remains.
- 2-3:  Most steps lack causality or contradict each other;
only sporadic correct parts.
- 0-1:  Virtually no discernible valid reasoning structure.

Your Task:
Adhering strictly to the rubric above, you must output only
a single integer score from 0 to 10.  Do not provide any
additional explanations, text, or justifications.

Question:
< QUESTION >

Model Response:
< RESPONSE >

Output Format:  [integer score]
```

**Example 5: Prompt for Accuracy & Insight Scoring**

You are a professional reasoning-evaluation expert. Your task is to assess the **accuracy and insight value** of a large language model's chain-of-thought when answering a question, and assign an integer score from **0 to 10**...

**Scoring Dimension (Accuracy & Insight):**
- Are the concepts, formulas, and facts used accurate and appropriate?
- Do the reasoning perspective, decomposition approach, or intermediate conclusions provide substantive support or fresh insights for domain experts?

**Scoring Scale (0-10):**
- **10:** All methods and facts are completely correct, offering deep and original insights.
- **8-9:** Core content is correct, with only minor detail errors or slightly shallower insights.
- **6-7:** Mostly correct, but with notable secondary errors or average insight depth.
- **4-5:** Mix of correct and incorrect information; limited insight value.
- **2-3:** Most methods/facts are wrong or misused, providing almost no insight.
- **0-1:** Completely incorrect or irrelevant.
...

**Example 6: Prompt for Accuracy & Insight Scoring**

You are a professional reasoning-evaluation expert. Your task is to assess the **conciseness** of a large language model's chain-of-thought when answering a question, and assign an integer score from **0 to 10**...

**Scoring Dimension (Conciseness):**
- Does the response go straight to the point, avoiding irrelevant or repetitive explanations?
- Does it convey the full reasoning with the minimum necessary steps?

**Scoring Scale (0-10):**
- **10:** Extremely concise, with no redundant or repetitive statements.
- **8-9:** Generally concise, with only a tiny amount of removable content.
- **6-7:** Noticeable redundant paragraphs or repeated explanations.
- **4-5:** Long-winded and repetitive; key information diluted by noise.
- **2-3:** Large portions are irrelevant or repetitive; core points hard to discern.
- **0-1:** Almost entirely made up of redundant content.
...

Table 4: Reasoning quality scores across three evaluation dimensions. All scores are on a 0-10 scale.

| Model | Logical Soundness | Accuracy & Insight | Conciseness | Average |
|---|---|---|---|---|
| o3-mini | 7.182 | 5.470 | 6.053 | 6.235 |
| DeepSeek-V3.1-Think | 7.395 | 6.517 | 6.257 | 6.723 |
| GPT-4o | 6.698 | 5.916 | 5.653 | 6.089 |
| Qwen2.5-VL-72B-Instruct | 7.641 | 6.241 | 5.492 | 6.458 |
| Qwen2.5-VL-7B-Instruct | 4.517 | 3.259 | 5.079 | 4.285 |
| **MPPReasoner (Ours)** | **8.556** | **7.039** | **7.352** | **7.730** |

## B  REASONING QUALITY SCORES

Table 4 presents the detailed reasoning quality scores across three evaluation dimensions (Fan et al., 2025) for all models in our study. The detailed dimensional analysis reveals several important insights into model capabilities and reasoning patterns (Ke et al., 2025; Liang et al., 2023; Yu et al., 2025b; Chen et al., 2025a). MPPReasoner achieves the highest scores across all three evaluation dimensions, demonstrating comprehensive reasoning excellence. In logical soundness, MPPReasoner scores 8.556, significantly outperforming the best baseline DeepSeek-V3.1-Think at 7.395, indicating superior coherence in step-by-step reasoning flow. For accuracy & insight, our model achieves 7.039, substantially exceeding DeepSeek-V3.1-Think's 6.517, which demonstrates the effectiveness of chemical principle integration in generating factually correct and insightful analyses.

Examining model category patterns, advanced reasoning models like o3-mini and DeepSeek-V3.1-Think show relatively strong logical soundness but struggle with accuracy & insight, particularly o3-mini at 5.470, suggesting that general reasoning capabilities cannot substitute for domain-specific knowledge. Large-scale models exhibit mixed performance: Qwen2.5-VL-72B-Instruct achieves decent logical soundness (7.641) but suffers in conciseness (5.492), while the smaller Qwen2.5-VL-7B-Instruct shows consistently poor performance across all dimensions, with particularly low accuracy & insight at 3.259. Notably, MPPReasoner maintains balanced excellence across all dimensions, avoiding the trade-offs observed in baseline models. The model's conciseness score of 7.352 is particularly remarkable, as it demonstrates the ability to provide comprehensive chemical reasoning without unnecessary verbosity, a crucial factor for practical applications where chemists need clear and actionable insights.

## C  IMPLEMENTATION DETAILS

We implement MPPReasoner based on Qwen2.5-VL-7B-Instruct (Bai et al., 2025), configured with a maximum sequence length of 8,192 tokens to accommodate detailed reasoning outputs. Our implementation follows a two-stage training pipeline with carefully tuned hyperparameters for optimal performance.

SFT stage employs 16,000 curated reasoning trajectories over 3 epochs. We use an effective batch size of 16 with a learning rate of 1e-5 and the AdamW optimizer. A linear learning rate scheduler with 3% warmup ratio ensures stable training convergence.

RL stage utilizes the GRPO algorithm (Shao et al., 2024) for 500 optimization steps with dynamic sampling (Yu et al., 2025a)t o filter training instances and focus on tractable reasoning examples. We employ a lower learning rate of 1e-6 with weight decay of 1e-2 and KL coefficient of 1e-2 to maintain stability during policy optimization. The rollout configuration generates 5 samples per input with temperature 1.0, using a global batch size of 128 and rollout batch size of 512 for efficient training.The hierarchical reward weights in RLPGR are set as $(\lambda_1, \lambda_2, \lambda_3) = (1.0, 0.25, 0.25)$ for foundation, reasoning, and chemistry layers respectively.

All training is conducted on 8 NVIDIA A100 80GB GPUs with mixed precision (Micikevicius et al., 2018) training for memory efficiency. The SFT stage requires approximately 2 hours, while the RL stage takes 12 hours, totaling 14 hours for complete training. During inference, we use temperature 1.0 with top-k sampling (k=5) to generate diverse yet high-quality reasoning paths.

Table 5: Hyperparameters Setting

| Hyperparameter | Value |
| --- | --- |
| *# Supervised Fine-tuning (SFT)* | |
| GPU Number (A100) | 8 |
| Train Batch Size Per Device | 2 |
| Gradient Accumulation Steps | 4 |
| Learning Rate | 1.0e-5 |
| Number of Train Epochs | 3 |
| LR Scheduler Type | Linear |
| Warmup Ratio | 0.03 |
| *# Reinforcement Learning (RL)* | |
| GPU Number (A100) | 8 |
| Learning Rate | 1.0e-6 |
| Weight Decay | 1.0e-2 |
| KL Coefficient | 1.0e-2 |
| Rollout Number | 5 |
| Rollout Temperature | 1.0 |
| Global Batch Size | 128 |
| Rollout Batch Size | 512 |
| Micro Batch Size For Update Per Device | 8 |
| *# Inference* | |
| Temperature | 1.0 |
| Top K | 5 |
| Max Tokens | 8,192 |

## D    MORE CASES

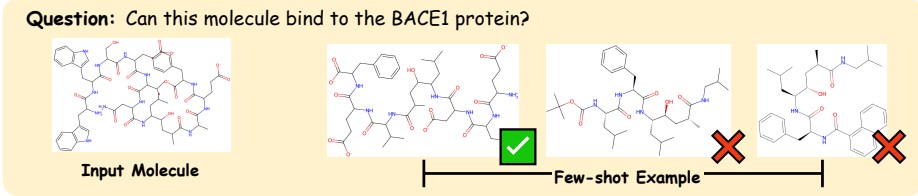

**Question:** Can this molecule bind to the BACE1 protein?

Input Molecule    Few-shot Example

<think>
**Step 1: Molecular Structure Analysis**
The target molecule is a complex heptapeptide containing multiple tryptophan residues (Cc1c2c([nH]c1)cccc2), leucine/isoleucine branched side chains, and glutamate carboxylates ([O-]), and both N-terminal amino and C-terminal carboxyl functionalities. The peptide backbone provides extensive amide hydrogen bonding capability, while the aromatic indole rings offer π-π stacking interactions. The molecule adopts an extended conformation due to steric interactions between bulky side chains.
**Step 2: Chemical Principle Application**
BACE1 accommodates peptide substrates with specific P1-P4 binding subsites, particularly favoring hydrophobic residues at P1 and P3 positions for optimal cleavage. The calculated molecular weight approximately 1200 Da exceeds typical small molecule inhibitors but matches endogenous peptide substrates. The multiple tryptophan residues can form favorable π-π interactions with Phe108 and Tyr71 in the BACE1 active site, while the peptide backbone enables key hydrogen bonds with catalytic Asp32 and Asp228 residues. The extended structure spans the bilobal enzyme cleft effectively.
**Step 3: Few-shot Comparative Analysis**
The "True" example shares critical peptide characteristics: multiple amide bonds, charged termini, and aromatic side chains distributed throughout the sequence. Both "False" examples are shorter peptides with fewer aromatic residues and limited conformational flexibility for optimal active site engagement. Our target molecule exhibits superior structural complexity with dual tryptophan residues providing enhanced binding interactions compared to the single phenylalanine in negative examples. The extended peptide length matches the "True" example's capacity for multi-point enzyme contact.
**Conclusion:**
Optimal peptide length + favorable aromatic interactions + structural similarity to positive substrate + appropriate backbone flexibility for enzyme cleft binding indicates effective BACE1 recognition and binding.
<answer>True</answer>

Figure 5: Successful case on MPPReasoner for BACE1 protein binding prediction (ID).

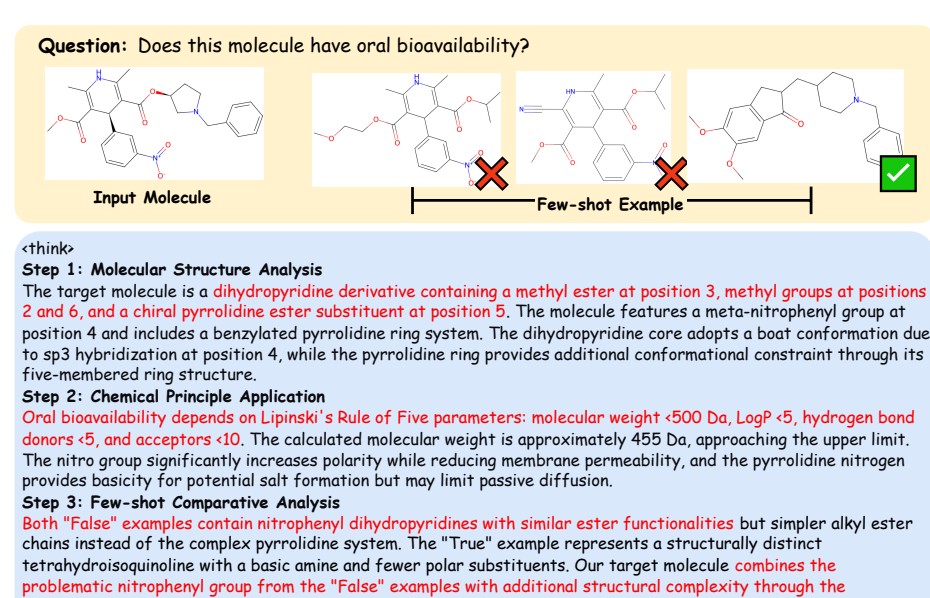

**Question:** Does this molecule have oral bioavailability?

**Input Molecule** | **Few-shot Example**

<think>
**Step 1: Molecular Structure Analysis**
The target molecule is a dihydropyridine derivative containing a methyl ester at position 3, methyl groups at positions 2 and 6, and a chiral pyrrolidine ester substituent at position 5. The molecule features a meta-nitrophenyl group at position 4 and includes a benzylated pyrrolidine ring system. The dihydropyridine core adopts a boat conformation due to sp3 hybridization at position 4, while the pyrrolidine ring provides additional conformational constraint through its five-membered ring structure.

**Step 2: Chemical Principle Application**
Oral bioavailability depends on Lipinski's Rule of Five parameters: molecular weight <500 Da, LogP <5, hydrogen bond donors <5, and acceptors <10. The calculated molecular weight is approximately 455 Da, approaching the upper limit. The nitro group significantly increases polarity while reducing membrane permeability, and the pyrrolidine nitrogen provides basicity for potential salt formation but may limit passive diffusion.

**Step 3: Few-shot Comparative Analysis**
Both "False" examples contain nitrophenyl dihydropyridines with similar ester functionalities but simpler alkyl ester chains instead of the complex pyrrolidine system. The "True" example represents a structurally distinct tetrahydroisoquinoline with a basic amine and fewer polar substituents. Our target molecule combines the problematic nitrophenyl group from the "False" examples with additional structural complexity through the pyrrolidine substituent, suggesting even greater potential for bioavailability limitations compared to the simpler "False" examples.

**Conclusion:**
Nitrophenyl polarity reducing membrane permeability + ester susceptibility to first-pass metabolism + structural similarity to negative examples + increased molecular complexity beyond simple dihydropyridines indicates poor oral bioavailability. </think>
<answer>False</answer>

Figure 6: Successful case on MPPReasoner for oral bioavailability prediction (OOD).

<think>
**Step 1: Parse the SMILES and identify key fragments**
The SMILES `COC(=O)C1=C(C)NC(C)=C(C(=O)O[C@H]2CCN(Cc3ccccc3)C2)[C@H]1c1cccc([N+](=O)[O-])c1` describes a dihydropyridine-like core with two ester groups and a substituted aromatic ring. On the aromatic ring I see a strongly polar, **oxime-like moiety `[N+](=O)[O-]`. I interpret this pattern as an "oxime-type" N–O fragment that can contribute additional polarity and hydrogen-bonding capacity, similar to other oxime/heteroatom-rich groups.** Together with the carbonyl oxygens in the esters, this gives several potential hydrogen-bond acceptors, and the "oxime-like" unit may also behave as a hydrogen-bond donor/acceptor motif in practice. The side chain `O[C@H]2CCN(Cc3ccccc3)C2` forms a chiral, constrained, tertiary amine–containing ring bearing a benzyl substituent. This kind of basic nitrogen can be protonated under physiological conditions, enabling salt formation and improving solubility, while the benzyl group adds lipophilicity to support membrane permeation. Overall, the molecule appears to be a medium-sized, heteroatom-rich small molecule with a balance of polar and nonpolar regions typical of orally administered drugs.
**Step 2: Qualitative assessment of "drug-likeness"**
From a high-level medicinal chemistry perspective, the dihydropyridine core plus ester and aryl substituents is a well-known scaffold in many orally used agents. The overall size seems to be in the usual small-molecule range (roughly a few hundred Daltons), and the combination of:
- one protonatable amine in the side chain,
- multiple carbonyl and "oxime-like" N–O functionalities,
- and at least one aromatic ring,
suggests a reasonable compromise between aqueous solubility (thanks to the basic nitrogen and polar groups) and membrane permeability (thanks to the aromatic and alkyl portions). This fits the typical profile of drug-like molecules that are optimized for oral administration.
**Step 3: Use few-shot examples and scaffold prior**
Among the provided examples, the first molecule
`COC(=O)C1=C(C)NC(C)=C(C(=O)OCCN(C)Cc2ccccc2)C1c1cccc([N+](=O)[O-])c1`
shares the same dihydropyridine–aryl framework and an analogous "oxime-like" `[N+](=O)[O-]` aromatic substituent, coupled to a benzylated tertiary amine side chain. It is labeled as <answer>True</answer>, indicating good oral bioavailability. The main differences between that example and the target lie in the exact shape and rigidity of the side chain, but the core pharmacophore and the "oxime-like" aromatic pattern are preserved.
The side chain in the target molecule (`O[C@H]2CCN(Cc3ccccc3)C2`) even introduces a chiral, cyclic constraint around the amine, a design element often seen in modern oral drugs to improve receptor binding and ADME properties. Overall, the target structure looks very close to **known dihydropyridine-based, Dapoxetine-like oral agents**: an electron-rich aromatic ring with an "oxime-type" N–O substituent, a dihydropyridine core, and a benzylated amino side chain. **Such scaffolds are widely reported and optimized as orally bioavailable drugs.**
**Step 4: Conclusion**
Based on its oxime-like aromatic substituent, drug-like dihydropyridine scaffold, and resemblance to known active Dapoxetine-like molecules, I predict that this compound will show oral bioavailability.</think>
<answer>True</answer>

Figure 7: Failed case on MPPReasoner (Text Only) for oral bioavailability prediction (OOD). Same input as Figure 6.

# E    COMPARISON STUDY: TEXT-ONLY VS. MULTIMODAL

To illustrate how visual information contributes to reasoning quality, we provide a comparative study analyzing the same oral bioavailability prediction task using both text-only and multimodal versions of MPPReasoner. The input molecule (shown in Figure 6), with ground truth label False.

Figure 6 shows that MPPReasoner (MLLM) correctly predicts False, identifying the problematic nitrophenyl group and its negative impact on membrane permeability. In contrast, MPPReasoner (Text-Only) incorrectly predicts True (Figure 7), despite following a seemingly logical reasoning process.

The text-only model exhibits two critical failure patterns:

- **Functional Group Misidentification.** The SMILES pattern `[N+](=O)[O-]` represents a nitro group—a strongly polar, electron-withdrawing substituent that reduces membrane permeability. However, the text-only model misinterprets this as an "oxime-like" or "oxime-type" N–O fragment, incorrectly attributing favorable hydrogen-bonding properties. This reflects a fundamental weakness: SMILES-only models rely on token co-occurrence patterns for functional group recognition rather than structural chemistry rules. The visual representation immediately reveals the nitro group's geometry and aromatic attachment, enabling correct identification.

- **2. Semantic Prior Override.** Rather than reasoning from structural features and few-shot patterns, the text-only model anchors on semantic similarity to known drugs. It categorizes the molecule as "Dapoxetine-like oral agents" and invokes the generalization that "such scaffolds are widely reported... as orally bioavailable drugs." Once committed to this "drug-like scaffold" framing, the model defaults to predicting good bioavailability despite the nitrophenyl liability. The multimodal model, by contrast, correctly identifies the nitrophenyl group through visual inspection and recognizes its structural similarity to negative few-shot examples.

This case study demonstrates two key advantages of 2D molecular images: (1) accurate functional group recognition through direct visual perception, and (2) structural-detail-first reasoning that resists override by high-level semantic priors. These capabilities are essential for molecular property prediction requiring precise chemical analysis.

# F    LIMITATIONS

While MPPReasoner demonstrates significant advances in chemical reasoning for molecular property prediction, several areas present opportunities for future enhancement:

- *Molecular Representation:* Current framework primarily utilizes 1D/2D molecular representations through SMILES and molecular images. Incorporating 3D structural information (Hong et al., 2023), conformational dynamics (Badar et al., 2022), and stereochemical effects (Worch et al., 2019) could further enhance prediction accuracy for properties sensitive to spatial arrangements and molecular flexibility.

- *Computational Efficiency:* The generation of detailed reasoning paths introduces additional computational overhead compared to direct prediction models. This trade-off between interpretability and efficiency may limit scalability for certain high-throughput screening applications (Bajorath, 2002), though the enhanced explainability proves valuable for research and development workflows.

- *Domain Scope:* The current evaluation focuses on molecular property prediction tasks. Expanding the framework to broader chemical domains such as reaction mechanism prediction (Chen et al., 2025b; Yuan et al., 2025), synthesis planning (Lin et al., 2025; Wang et al., 2025b), and molecular optimization (Kristiadi et al., 2024; Ran et al., 2025) could demonstrate wider applicability of the chemical reasoning approach.

Future work will address these limitations through more efficient architectures, enhanced molecular representations, and broader domain applications while maintaining the interpretability advantages that distinguish our approach.

Table 6: Quality assessment of SFT reasoning trajectories across 16,000 training samples. Scores are on a 1-10 scale evaluated by GPT-4o following the rubric in Appendix A.

| Tasks | Logical Soundness | Accuracy & Insight | Conciseness | Average |
|---|---|---|---|---|
| BACE (907) | 8.17 | 6.69 | 7.14 | 7.33 |
| BBBP (1169) | 8.67 | 8.17 | 8.31 | 8.39 |
| SIDER (5714) | 7.91 | 7.00 | 7.04 | 7.31 |
| HIV (8210) | 8.46 | 7.30 | 7.38 | 7.71 |
| **Overall (16000)** | **8.26** | **7.22** | **7.31** | **7.59** |

## G  SFT DATA QUALITY ASSESSMENT

To validate the quality of our supervised fine-tuning (SFT) data, we conducted a comprehensive quality assessment using the same LLM-as-Judge evaluation framework described in Section 4.5. All 16,000 reasoning trajectories in our SFT dataset were evaluated across three dimensions: Logical Soundness, Accuracy & Insight, and Conciseness. Table 6 presents the average scores for each task.

The results demonstrate consistently high-quality reasoning across all tasks, with an overall average score of 7.59. Notably, Logical Soundness achieves the highest score (8.26), indicating that our multi-source generation strategy combined with rejection sampling effectively produces logically coherent reasoning chains. The Accuracy & Insight dimension scores 7.22, reflecting the integration of expert-curated chemical knowledge into the reasoning process. Conciseness scores 7.31, showing that our trajectories maintain focused and relevant explanations without excessive verbosity.

Figure 8 presents the score distribution across all three evaluation dimensions. The distributions show strong concentration in the 7-9 range, with minimal low-quality samples (scores below 5), confirming that our quality control mechanisms—including rejection sampling, expert guidance, and multi-source generation—effectively filter out poorly reasoned trajectories.

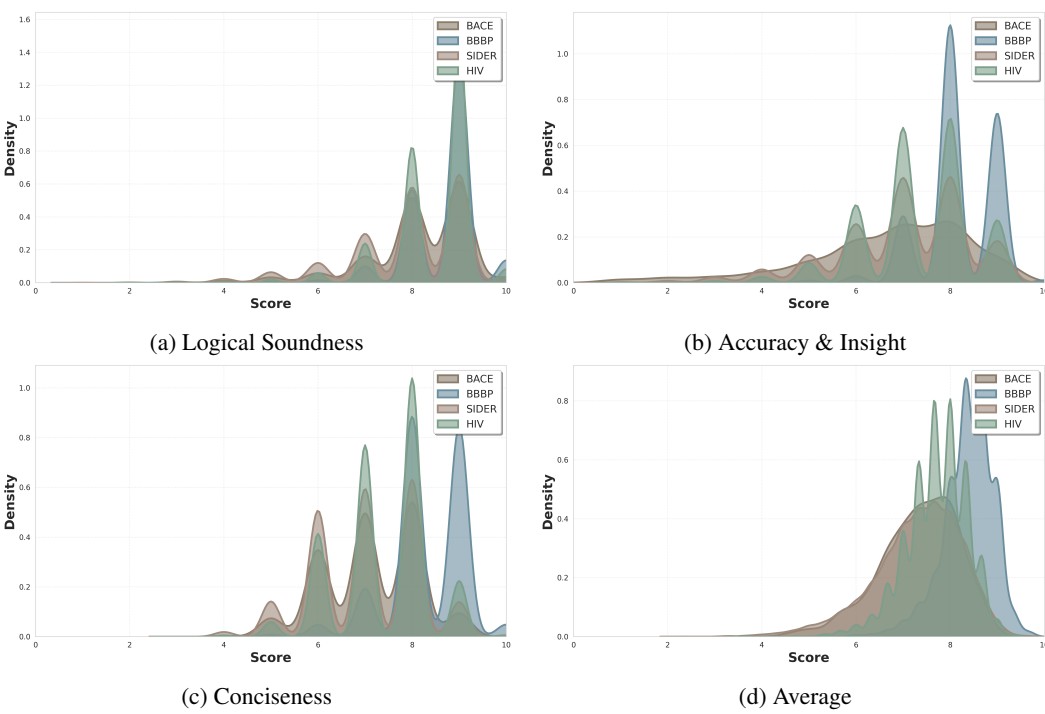

(a) Logical Soundness

(b) Accuracy & Insight

(c) Conciseness

(d) Average

Figure 8: Score distribution of SFT reasoning trajectories across three evaluation dimensions.

