# OpenReview forum: "Reasoning-Enhanced Large Language Models for Molecular Property Prediction"
_ICLR.cc/2026/Conference — Submitted to ICLR 2026_

### Official Review · Reviewer_XoHL · 2025-10-27

**Soundness:** 2
**Presentation:** 3
**Contribution:** 2
**Rating:** 4
**Confidence:** 3

**Summary:**

This submission focuses on the molecular property prediction problem. Prior methods exhibit poor cross-task generalization and a lack of interpretability. The author argues that the fundamental limitation shared by previous methods is the absence of effective chemical reasoning, where models need to analyze the molecule and explain the prediction under chemical principles. To address this limitation and introduce chemical reasoning to the model, the submission proposes MPPReasoner, a reinforcement learning (RL) approach to instill domain-specific reasoning ability to the multi-modal large language model (MLLMs). MPPReasoner consists of two training stages: supervised fine-tuning (SFT) for instilling reasoning ability to the model, and Reinforcement Learning from Principle-Guided Rewards (RLPGR) for further enhancing the chemical reasoning ability. The submission conducts extensive empirical studies and justifies the effectiveness of the proposed method.

**Strengths:**

- The submission focuses on the trendy LLM reasoning problem, effectively enhancing the model's reasoning ability on the molecular property prediction task by proposing a domain-specific reward to guide the training process.
- The submission is generally well-written, with clear illustrations and tables.
- Extensive experiments have been conducted to provide a good insight into the components of the proposed method.

**Weaknesses:**

- The overall pipeline is similar to the existing framework [1], where the model is first SFT with distilled reasoning trajectories and followed by RLVR. The proposed RLPGR differs by employing domain-specific rule checks via RDKit. Further experiments are suggested to investigate the underlying effectiveness of the chemistry layer. For example, whether the reward truly contributes to the quality and chemical validity of the reasoning trajectories.
- The quality control of the reasoning trajectories for SFT could be further enhanced beyond rejection sampling. Since the submission emphasises the chemical reasoning ability, the data for SFT could be checked by the LLM-as-Judge to ensure the reasoning quality. In addition, the reward definitions are not clear. For example, the reasoning layer proposes to evaluate the reasoning quality from logic consistency and a comparative perspective. However, neither of which are not provides implementation details, such as the adopted keyword for the logical consistency reward.
- The rationale for adopting an MLLM instead of a purely textual LLM requires further justification. Since SMILES strings already encode molecular structure information, the claimed benefit of incorporating 2D molecular images is not convincingly supported by experiments. Moreover, most domain-specific baselines (e.g., BioT5-Plus, MolecularGPT) operate purely on textual modalities, making the comparison potentially unfair. The authors should provide ablations or control experiments isolating the contribution of the visual modality.

[1] DeepSeek-R1 incentivizes reasoning in LLMs through reinforcement learning. In Nature, 2025.

**Questions:**

1. Could you provide further discussion on the adaptation of MLLM instead of a purely textual LLM?
2. Could you provide discussion or experiments on the reasoning quality of the SFT data?
3. Could you provide more detailed information on the reasoning layer in Section 3.2.2?

---

> ### Author Response · Authors · 2025-11-26
> **Response to Reviewer XoHL (Part 1/3)**
>
> > W1: The overall pipeline is similar to the existing framework [1], where the model is first SFT with distilled reasoning trajectories and followed by RLVR. The proposed RLPGR differs by employing domain-specific rule checks via RDKit.  Further experiments are suggested to investigate the underlying effectiveness of the chemistry layer.  For example, whether the reward truly contributes to the quality and chemical validity of the reasoning trajectories.
>
> **Response to (W1)**: We appreciate the reviewer's observation and the opportunity to clarify the distinctions between our work and DeepSeek-R1 [1]. While both employ SFT followed by RL, our contributions extend significantly beyond adapting this general pipeline to molecular property prediction.
>
> 1. **Distinctions from DeepSeek-R1**
>
> Our work differs from DeepSeek-R1 [1] in several fundamental aspects:
>
> - **Domain-Specific Reasoning Verification vs. General Reasoning**: DeepSeek-R1 focuses on incentivizing general reasoning capabilities through process rewards and outcome supervision. In contrast, **RLPGR introduces a systematic framework for domain-specific scientific reasoning verification** through computational validation of chemical principles, representing a paradigm shift from general to scientific reasoning.
>
> - **Computational Verification vs. Learned Rewards**: While DeepSeek-R1 relies on learned reward models or human preference data, **RLPGR employs computational verification via RDKit** to provide objective, verifiable feedback on chemical correctness. This eliminates subjective bias and ensures scientific accuracy through molecular property calculations and substructure detection.
>
> - **Hierarchical Reward Decomposition**: Our three-layer reward structure (Foundation/Reasoning/Chemistry) systematically decomposes scientific reasoning quality into verifiable components, specifically designed for domains where correctness can be computationally validated—fundamentally different from general preference-based rewards.
>
> 2. **Evidence of Chemistry Layer Effectiveness**
>
> Regarding the reviewer's question about whether the Chemistry Layer rewards truly contribute to reasoning quality and chemical validity, we provide multiple lines of evidence:
>
> **Quantitative Evidence (Table 2):**
> The ablation study demonstrates that adding the Chemistry Layer (chemical principle + structural analysis rewards) provides **substantial incremental improvements**:
> - **+3.92% on ID tasks** (from 0.7819 to 0.8190)
> - **+2.61% on OOD tasks** (from 0.7780 to 0.7977)
>
> **Qualitative Evidence (Section 5 & Appendix D):**
> Our case studies provide concrete examples of how RLPGR training—particularly the Chemistry Layer—improves reasoning quality:
>
> - **Figure 4** presents a direct comparison between GPT-4o and MPPReasoner on CYP450-2C9 substrate prediction, demonstrating how our model produces chemically accurate reasoning while GPT-4o exhibits:
>   - **Structural analysis errors**: Incorrectly assuming amide groups enhance binding affinity
>   - **Overgeneralization**: Broadly claiming nitrogen-containing compounds show P450 compatibility
>   - **Unsupported reasoning**: Making statistical generalizations without chemical basis
>
> - **Appendix D (Figures 5-6)** provides additional successful cases showing systematic chemical reasoning with:
>   - Accurate functional group identification verified by RDKit
>   - Correct chemical principle application (e.g., LogP calculations, BACE1 binding requirements)
>   - Precise structural analysis aligned with molecular properties
>
> **Reasoning Quality Assessment (Table 3):**
> The Chemistry Layer's effectiveness is further validated through reasoning quality evaluation, where MPPReasoner achieves **7.039 in Accuracy & Insight** compared to the best baseline at 6.517. This dimension specifically measures chemical concept correctness and structural analysis accuracy, directly reflecting the Chemistry Layer's contribution.
>
> ---
>
> [1] Guo et al., DeepSeek-R1: Incentivizing Reasoning in LLMs Through Reinforcement Learning, Nature, 2025.

---

> ### Author Response · Authors · 2025-11-26
> **Response to Reviewer XoHL (Part 2/3)**
>
> > W2: The quality control of the reasoning trajectories for SFT could be further enhanced beyond rejection sampling. Since the submission emphasises the chemical reasoning ability, the data for SFT could be checked by the LLM-as-Judge to ensure the reasoning quality. In addition, the reward definitions are not clear. For example, the reasoning layer proposes to evaluate the reasoning quality from logic consistency and a comparative perspective. However, neither of which are not provides implementation details, such as the adopted keyword for the logical consistency reward.
>
> > Q2: Could you provide discussion or experiments on the reasoning quality of the SFT data?
>
> **Response to (W2 & Q2)**: We thank the reviewer for these constructive suggestions on enhancing quality control and providing clearer implementation details.
>
> 1. **SFT Data Quality Assessment**
>
> We appreciate the reviewer's suggestion to evaluate SFT data quality using LLM-as-Judge. We have conducted comprehensive quality assessment experiments on our SFT reasoning trajectories using the same evaluation framework described in Section 4.5. The results demonstrate high-quality reasoning patterns across all three dimensions:
> | Tasks | Logical Soundness | Accuracy & Insight | Conciseness | Average |
> |-------|-------------------|-------------------|-------------|---------|
> | BACE (907) | 8.17 | 6.69 | 7.14 | 7.33 |
> | BBBP (1169) | 8.67 | 8.17 | 8.31 | 8.39 |
> | SIDER (5714) | 7.91 | 7.00 | 7.04 | 7.31 |
> | HIV (8210) | 8.46 | 7.30 | 7.38 | 7.71 |
> | Avg. (16000) | 8.26 | 7.22 | 7.31 | 7.59 |
>
> The average reasoning quality score of 7.59 across 16,000 training trajectories confirms that our multi-layered quality control approach (rejection sampling + expert guidance + multi-source generation) produces high-quality training data that establishes a strong foundation for chemical reasoning capabilities. Detailed score distributions are provided in Appendix G.
>
> 2. **Detailed Reward Implementation**
>
> Regarding the clarity of reward definitions, we acknowledge that the implementation details were insufficiently detailed in the original submission. We have now provided comprehensive technical specifications in our response to **Q3**, which includes:
>
> - **Logical Consistency Reward ($r_{cons}$)**: Detailed explanation of the two-step verification process using weighing/conclusion keywords (e.g., "therefore", "in conclusion", "overall") and sentiment confirmation terms for True/False predictions.
> - **Comparative Analysis Reward ($r_{comp}$)**: Implementation details showing how the system detects explicit mentions of few-shot example SMILES strings within reasoning text.
> - **Complete keyword sets and verification logic**: Will be provided Supplementary Materials and Anonymous Repository.
>
> We will incorporate these implementation details into the revised manuscript (Section 3.2.2) to ensure full reproducibility and clarity.
>
> ---
>
> > W3: The rationale for adopting an MLLM instead of a purely textual LLM requires further justification. Since SMILES strings already encode molecular structure information, the claimed benefit of incorporating 2D molecular images is not convincingly supported by experiments. Moreover, most domain-specific baselines (e.g., BioT5-Plus, MolecularGPT) operate purely on textual modalities, making the comparison potentially unfair. The authors should provide ablations or control experiments isolating the contribution of the visual modality.
>
> > Q1: Could you provide further discussion on the adaptation of MLLM instead of a purely textual LLM?
>
> **Response to (W3 & Q1)**: We sincerely appreciate the reviewer's concern about justifying our MLLM architecture choice. In response, we have conducted comprehensive ablation experiments and control studies to isolate the contribution of the visual modality. Please refer to the **General Response: Multimodal Ablation Study and Justification** for detailed experimental analysis.

---

> > ### Author Response · Authors · 2025-11-26
> > **Response to Reviewer XoHL (Part 3/3)**
> >
> > > Q3: Could you provide more detailed information on the reasoning layer in Section 3.2.2?
> >
> > **Response to (Q3)**: We thank the reviewer for requesting clarification on the Reasoning Layer implementation. We will provide more detailed technical information in the revised manuscript. The Reasoning Layer evaluates general reasoning quality through two complementary components designed to ensure logical coherence and effective use of contextual information:
> >
> > **1. Logical Consistency Reward ($r_{cons}$)**
> >
> > This reward measures alignment between the reasoning conclusion and the final prediction by analyzing sentiment consistency through a two-step verification process:
> >
> > First, the system detects whether the reasoning contains **weighing or conclusion keywords** (e.g., "therefore", "in conclusion", "overall", "consequently", "taking all into account"), which indicate that the model is making an explicit final judgment.
> >
> > Second, it verifies consistency between the reasoning sentiment and the prediction:
> > - For **True predictions**: checks for affirmative confirmation terms (e.g., "likely true", "supports the property", "stronger case", "consequently is")
> > - For **False predictions**: checks for negative confirmation terms (e.g., "likely false", "lacks the property", "counter-argument", "consequently is not")
> > - Awards $r_{cons} = 1$ if both conditions are satisfied, otherwise $r_{cons} = 0$
> >
> > This mechanism ensures that the reasoning process logically supports the final answer through explicit conclusion statements, preventing cases where correct predictions are made despite contradictory or absent reasoning conclusions.
> >
> > **2. Comparative Analysis Reward ($r_{comp}$)**
> >
> > This reward encourages the model to effectively utilize few-shot examples retrieved based on molecular similarity. The implementation:
> > - Searches for **explicit mentions of the few-shot example SMILES strings** within the reasoning text
> > - Awards $r_{comp} = 1$ if **at least one example is explicitly referenced**
> > - Otherwise: $r_{comp} = 0$
> >
> > This mechanism promotes analogical reasoning capabilities by incentivizing the model to explicitly reference and compare with similar molecules in the few-shot context, enhancing cross-molecular reasoning and generalization.
> >
> > We will expand Section 3.2.2 in the revised manuscript to include these implementation details, and provide complete keyword sets and verification logic in the Supplementary Materials to ensure full reproducibility.

---

### Official Review · Reviewer_6cSd · 2025-10-31

**Soundness:** 2
**Presentation:** 1
**Contribution:** 1
**Rating:** 2
**Confidence:** 4

**Summary:**

This paper proposes MPPReasoner, a framework for learning chemical reasoning in molecular property prediction. Specifically, it tackles the fact that existing models, such as GNNs or molecular LLMs, are limited in providing a chemical reasoning process while predicting. The proposed method uses a two-staged training pipeline. First, the model is supervised fine-tuned (SFT) on the ChemCoT dataset using curated instructions that embed task-specific knowledge. Second, it is further finetuned with GPRO using a composite reward that combines foundational rewards (answer accuracy and format compliance), reasoning rewards (logical consistency and use of few-shot examples), and chemical rewards (chemical concept correctness and structural analysis accuracy). The experimental results show that the proposed training algorithm outperforms task-specific specialists and LLM-based generalists on both in-distribution and OOD tasks. Further ablation study demonstrates the contribution of each reward component.

**Strengths:**

* The proposed reward is designed for comprehensively assessing the chemical reasoning process.
* The ablation study shows the effectiveness of each reward component.

**Weaknesses:**

* My major concern is the novelty. Sequentially training SFT and RL is a well-known technique for fine-tuning LLMs, and the constructed dataset is borrowed from prior work (ChemCoT). The novel aspect of this paper is the design of the reward function, but its contribution is limited.

* The claimed mitigation of limitations in existing approaches is not sufficiently persuasive. Recent molecular LLMs [1,2,3,4] provide informative reasoning processes. If such models demonstrate interpretability, the necessity of the proposed training algorithm should be clearly articulated. I recommend revising the overall motivation so that the limitations being addressed are aligned with the proposed reward design. For example, it would be better to present an analysis of whether the reasoning processes are well aligned with the final predictions or whether the predicted chemical concepts are correct, and then explain the motivation for the reward design on that basis.

* The general-purpose baselines are limited to GPT, DeepSeek, and the base model (e.g., Qwen2.5-VL). I suggest comparing against a broader range of models such as GPT-5, Qwen3, Gemma, and other fine-tuned multimodal LLMs [1,2,3,4].

* It is unclear why large vision–language models (LVLMs) are used as the baseline. Molecules can be represented in diverse forms such as text (e.g., SMILES) and molecular graphs. Nevertheless, the paper does not clearly justify why the proposed model needs to be built on top of an LVLM.

[1] Li et al., "Towards 3d molecule-text interpretation in language models." ICLR, 2024.

[2] Park et al., "LLaMo: Large Language Model-based Molecular Graph Assistant", NeurIPS, 2024.

[3] Kim et al., "Mol-LLaMA: Towards General Understanding of Molecules in Large Molecular Language Model", ArXiv, 2025.

[4] Wang et al., "TxGemma: Efficient and Agentic LLMs for Therapeutics", ArXiv, 2025.

**Questions:**

* Could the authors report the experimental results with greedy sampling (i.e., temperature = 0)
* Could the authors provide the experimental results on other CYP datasets such as CYP 3A4, 1A2, and 2C9?

---

> ### Author Response · Authors · 2025-11-26
> **Response to Reviewer 6cSd (Part 1/4)**
>
> > W1: My major concern is the novelty. Sequentially training SFT and RL is a well-known technique for fine-tuning LLMs, and the constructed dataset is borrowed from prior work (ChemCoT). The novel aspect of this paper is the design of the reward function, but its contribution is limited.
>
> **Response to (W1)**: We appreciate the reviewer's feedback and acknowledge the need to better articulate our contributions. However, we respectfully disagree with the characterization that our novelty is limited. Our work introduces several significant innovations beyond standard SFT+RL pipelines and reward function design.
>
> 1. **Core Contributions: Beyond Standard SFT+RL**
>
> Our primary contribution is not simply applying SFT+RL, but rather introducing **a systematic chemical reasoning framework** that enables LLMs to perform domain-specific scientific reasoning with verifiable quality assurance. This represents a fundamental shift from general-purpose reasoning to principle-guided chemical analysis.
>
> 2. **RLPGR: Domain-Specific Reasoning Verification System**
>
> While the reviewer characterizes our contribution as "reward function design," RLPGR is fundamentally **a domain-specific reasoning verification system** rather than a simple reward function. The key innovations include:
>
> - **Computational Verification via RDKit**: Unlike traditional RLHF that relies on human preferences or learned reward models, RLPGR leverages **computational verification** to provide objective, scalable feedback on chemical reasoning quality. This eliminates human annotation requirements while ensuring scientific accuracy through verifiable molecular property calculations and substructure detection.
> - **Hierarchical Reward Decomposition**: Our three-layer reward structure (Foundation/Reasoning/Chemistry) systematically decomposes the complex cognitive process of chemical reasoning into measurable, verifiable components. This hierarchical design is specifically tailored for scientific reasoning tasks where domain expertise can be computationally validated, distinguishing it from general RLHF approaches.
> - **Principle-Guided Rewards**: Unlike traditional RLHF that optimizes for human preference alignment, RLPGR evaluates adherence to established chemical principles through computational tools. This makes it particularly suitable for scientific reasoning where objective correctness can be verified, representing a paradigm shift from preference-based to principle-based reinforcement learning.
>
> 3. **Data Construction: Beyond ChemCoT Format**
>
> Regarding the ChemCoT reference, we want to clarify that we **only adopt the reasoning format structure from ChemCoT, not the dataset itself**. Our contribution includes:
>
> - **16,000 newly constructed reasoning trajectories** generated through our expert-guided multi-source teacher model approach
> - **Expert knowledge integration**: Curated domain knowledge from 5 chemists with cross-validation, providing systematic guidance for reasoning generation
> - **Multi-source synthesis strategy**: Parallel use of three state-of-the-art models (GPT-4o, DeepSeek-v3.1, Qwen2.5-VL) to ensure diversity and quality
>
> This data construction methodology represents a novel approach to generating domain-specific reasoning trajectories at scale, complementing the ChemCoT format with systematic expert knowledge integration.

---

> > ### Author Response · Authors · 2025-11-26
> > **Response to Reviewer 6cSd (Part 2/4)**
> >
> > > W2: The claimed mitigation of limitations in existing approaches is not sufficiently persuasive. Recent molecular LLMs [1,2,3,4] provide informative reasoning processes. If such models demonstrate interpretability, the necessity of the proposed training algorithm should be clearly articulated. I recommend revising the overall motivation so that the limitations being addressed are aligned with the proposed reward design. For example, it would be better to present an analysis of whether the reasoning processes are well aligned with the final predictions or whether the predicted chemical concepts are correct, and then explain the motivation for the reward design on that basis.
> >
> > **Response to (W2)**: We sincerely thank the reviewer for this insightful comment. We acknowledge that our motivation was not articulated as precisely as it should have been, and we appreciate the opportunity to clarify our contribution relative to existing molecular LLMs.
> >
> > 1. **Refined Motivation: From Interpretability to Reasoning Quality**
> >
> > The reviewer is correct that recent molecular LLMs [1,2,3,4] provide interpretable reasoning processes. We should not have emphasized interpretability as our primary differentiator. Instead, **our core motivation is to address the quality and verifiability of chemical reasoning**, which remains a critical challenge even when reasoning processes are interpretable.
> >
> > While existing approaches [1,2,3,4] generate interpretable reasoning, they lack systematic mechanisms to ensure:
> >
> > 1. **Reasoning-Prediction Alignment**: Whether the reasoning process logically supports the final prediction.
> > 2. **Chemical Principle Correctness**: Whether mentioned chemical concepts (e.g., LogP, hydrogen bonding, steric hindrance) are correctly applied and computationally verifiable.
> > 3. **Structural Analysis Accuracy**: Whether identified functional groups and structural features actually exist in the molecule.
> >
> > These limitations stem from training without domain-specific quality verification mechanisms. Models may learn to generate plausible-sounding reasoning without ensuring chemical correctness or logical consistency.
> >
> > 2. **Our Solution: Two-Stage Quality Assurance**
> >
> > MPPReasoner addresses these limitations through a two-stage training strategy specifically designed for reasoning quality:
> >
> > - **Stage 1 (SFT)**: We employ expert-guided data synthesis with quality control mechanisms—rejection sampling and multi-source teacher models. This establishes a high-quality foundation for chemical reasoning.
> >
> > - **Stage 2 (RLPGR)**: We systematically refine reasoning quality through hierarchical rewards that computationally verify: (1) reasoning-prediction alignment through logical consistency checks, (2) chemical principle correctness via RDKit-based molecular property validation, and (3) structural analysis accuracy through substructure detection. This computational verification enables systematic quality refinement beyond what supervised learning alone can achieve.
> >
> > We appreciate the reviewer's guidance on refining our motivation. Following this discussion, we will revise Section 1 (Introduction) to clearly articulate how our work addresses reasoning quality verification as a distinct challenge from interpretability, and how RLPGR's design directly targets the specific limitations identified above. We welcome any further feedback on how to best position this contribution.
> >
> > ---
> >
> > [1] Li et al., "Towards 3d molecule-text interpretation in language models." ICLR, 2024.\
> > [2] Park et al., "LLaMo: Large Language Model-based Molecular Graph Assistant", NeurIPS, 2024.\
> > [3] Kim et al., "Mol-LLaMA: Towards General Understanding of Molecules in Large Molecular Language Model", ArXiv, 2025.\
> > [4] Wang et al., "TxGemma: Efficient and Agentic LLMs for Therapeutics", ArXiv, 2025.

---

> > > ### Author Response · Authors · 2025-11-26
> > > **Response to Reviewer 6cSd (Part 3/4)**
> > >
> > > > W3: The general-purpose baselines are limited to GPT, DeepSeek, and the base model (e.g., Qwen2.5-VL). I suggest comparing against a broader range of models such as GPT-5, Qwen3, Gemma, and other fine-tuned multimodal LLMs [1,2,3,4].
> > >
> > > **Response to (W3)**:
> > > We sincerely thank the reviewer for this constructive suggestion. We have conducted additional experiments with a broader range of baseline models to provide more comprehensive evaluation.
> > >
> > > We have now evaluated the following additional models under the same few-shot experimental settings:
> > >
> > > - **General LLM**: Qwen3 Series[5](Qwen3-8B, Qwen3-VL-8B-Instruct, Qwen3-VL-8B-Thinking), Gemma-3-12b-it[6]
> > > - **Molecular LLMs**: 3D-MoLM [1], Mol-LLaMA [3], TxGemma [4]
> > >
> > > We were unable to include LLaMo [2] as critical data preprocessing steps were not released, making it difficult to adapt to our tasks. GPT-5 [7] was also excluded because it does not provide access to model logprobs (https://community.openai.com/t/logprobs-deprecated-for-gpt-5-models/1355427/2), which are necessary for computing ROC-AUC metrics.
> > >
> > > The extended results are presented in the updated Table 1:
> > >
> > > | Model | BACE | BBBP | SIDER | HIV | Bioavail. | C2C9_V | C2D6_V | AMES | ID Avg. | OOD Avg. |
> > > |-------|------|------|-------|-----|-----------|--------|--------|------|---------|----------|
> > > | 3D-MoLM | 0.7287 | 0.5141 | 0.5073 | 0.6603 | 0.6066 | 0.7081 | 0.7029 | 0.7190 | 0.6026 | 0.6842 |
> > > | Mol-LLaMA | 0.8349 | 0.6263 | 0.5576 | 0.7249 | 0.6020 | 0.7556 | 0.7789 | 0.7928 | 0.6859 | 0.7323 |
> > > | TxGemma | 0.6380 | 0.7102 | 0.5619 | 0.5235 | 0.5813 | 0.8075 | 0.7595 | 0.7907 | 0.6084 | 0.7348 |
> > > | Gemma-3-12b-it | 0.8526 | 0.6802 | 0.5652 | 0.6886 | 0.6194 | 0.7886 | 0.7780 | 0.8130 | 0.6967 | 0.7498 |
> > > | Qwen3-8B | 0.7924 | 0.6961 | 0.6442 | 0.5706 | 0.6181 | 0.7496 | 0.7165 | 0.8392 | 0.6758 | 0.7309 |
> > > | Qwen3-VL-8B-Instruct | 0.8757 | 0.7096 | 0.5905 | 0.7143 | 0.5919 | 0.8012 | 0.7784 | 0.8514 | 0.7225 | 0.7557 |
> > > | Qwen3-VL-8B-Thinking | 0.8597 | 0.7055 | 0.5973 | 0.5890 | 0.6073 | 0.7883 | 0.7572 | 0.8718 | 0.6879 | 0.7562 |
> > > | **MPPReasoner (Ours)** | **0.9090** | **0.7436** | **0.8280** | **0.7932** | **0.6728** | **0.8480** | **0.7950** | **0.8750** | **0.8190** | **0.7977** |
> > >
> > > MPPReasoner achieves the best performance across all datasets, demonstrating robust advantages over both general-purpose LLMs (Qwen3 series, Gemma) and specialized molecular LLMs (3D-MoLM, Mol-LLaMA, TxGemma).
> > >
> > > MPPReasoner outperforms the best baseline by **13.4%** on average ID performance, with particularly strong gains on challenging tasks like SIDER (+40.1% over best baseline). Our method achieves **5.5%** improvement over the best baseline on OOD tasks, validating that our chemical reasoning framework enhances cross-task generalization. Despite being specialized for molecular tasks, models like 3D-MoLM, Mol-LLaMA, and TxGemma show substantially lower performance than MPPReasoner, highlighting the effectiveness of our systematic reasoning framework and RLPGR training.
> > >
> > > These comprehensive comparisons confirm that MPPReasoner's performance gains are robust across a diverse range of competing approaches, validating the effectiveness of our chemical reasoning framework. We will include these extended results in the revised Table 1.
> > >
> > > ---
> > >
> > > > W4: It is unclear why large vision–language models (LVLMs) are used as the baseline. Molecules can be represented in diverse forms such as text (e.g., SMILES) and molecular graphs. Nevertheless, the paper does not clearly justify why the proposed model needs to be built on top of an LVLM.
> > >
> > > **Response to (W4)**: We thank the reviewer for raising this important architectural question. To directly address this concern, we have conducted systematic ablation experiments comparing text-only LLM and multimodal LVLM implementations of our framework. Please see the **General Response: Multimodal Ablation Study and Justification** for detailed experimental results.
> > >
> > >
> > > ---
> > >
> > > [1] Li et al., "3D-MOLM: Towards 3d molecule-text interpretation in language models", ICLR 2024.\
> > > [2] Park et al., "LLaMo: Large Language Model-based Molecular Graph Assistant", NeurIPS 2024.\
> > > [3] Kim et al., "Mol-LLaMA: Towards General Understanding of Molecules in Large Molecular Language Model", NIPS 2025.\
> > > [4] Wang et al., "TxGemma: Efficient and Agentic LLMs for Therapeutics", arXiv 2025.\
> > > [5] Bai et al., "Qwen3 Technical Report", arXiv 2025.\
> > > [6] Kamath et al., "Gemma 3 Technical Report", ArXiv, 2025.\
> > > [7] OpenAI, "GPT-5 System Card", 2025.

---

> > > > ### Author Response · Authors · 2025-11-26
> > > > **Response to Reviewer 6cSd (Part 4/4)**
> > > >
> > > > > Q2: Could the authors provide the experimental results on other CYP datasets such as CYP 3A4, 1A2, and 2C9?
> > > >
> > > > **Response to (Q2)**:
> > > > We thank the reviewer for this suggestion. We note that CYP2C9_veith is already included in our OOD benchmark (reported as C2C9_V in Table 1). We have conducted additional experiments on CYP1A2 and CYP3A4 datasets from the TDC benchmark to provide a more comprehensive evaluation of our model's performance.
> > > >
> > > > | Model | C1A2_veith | C3A4_veith |
> > > > |-------|------------|------------|
> > > > | 3D-MoLM | 0.7051 | 0.6447 |
> > > > | Mol-LLaMA | 0.8362 | 0.7295 |
> > > > | TxGemma (Trained on TDC) | 0.8672 | 0.8116 |
> > > > | Gemma-3-12b-it | 0.8159 | 0.7794 |
> > > > | GPT4o | 0.8239 | 0.7889 |
> > > > | Qwen3-VL-8B-Thinking | 0.8577 | 0.7822 |
> > > > | Qwen2.5-VL-7B-Instruct | 0.5257 | 0.5830 |
> > > > | **MPPReasoner** | **0.8768** | **0.8121** |
> > > >
> > > > - MPPReasoner achieves significant performance gains compared to its base model (Qwen2.5-VL-7B-Instruct), with improvements of **+66.8% on CYP1A2** and **+39.3% on CYP3A4**, demonstrating the effectiveness of our chemical reasoning framework on these additional CYP tasks.
> > > >
> > > > - Notably, **TxGemma was trained on TDC datasets**, meaning CYP1A2 and CYP3A4 are in-distribution tasks for this model. In contrast, MPPReasoner was not trained on any TDC data, making these genuinely out-of-distribution tasks for our model. Despite this disadvantage, MPPReasoner achieves **comparable performance to TxGemma**, and **outperforms all other baseline models** including specialized molecular LLMs and large-scale general models.

---

### Official Review · Reviewer_vgEK · 2025-10-31

**Soundness:** 1
**Presentation:** 3
**Contribution:** 3
**Rating:** 6
**Confidence:** 3

**Summary:**

Summary:
This paper introduces MPPReasoner, a multimoda large language model designed to predict molecular properties by incorporating chemical reasoning. The model is trained using a two-stage strategy involving supervised fine-tuning on expert-generated reasoning trajectories and a novel reinforcement learning method called RLPGR, which use rule-based rewards to evaluate the model's application of chemical principle. Experiments show that MPPReasoner outperforms existing models, particularly on out-of-distribution tasks, and generates chemically sound, interpretable explanations for its predictions.

**Strengths:**

Pros:
- The paper introduces MPPReasoner, a multimodal LLM for predicting molecular properties with chemical reasoning.
- The model achieved better perfromance compared with baseslines, especially on out-of-distribution tasks.

**Weaknesses:**

Cons:
- The framework primarily relies on SMILES and 2D molecular data. What if we want to consider the 3D molecular structural information as well?
- Ablation study on whether using 2D images helps is encouraged to be conducted.
- Computational cost may be an issue. Generating detailed, step-by-step reasoning paths creates more computational overhead than models that make direct predictions.
- There is also a quite related study that is worth discussing in the paper [1].

[1] Zheng, Y., Koh, H. Y., Ju, J., Nguyen, A. T., May, L. T., Webb, G. I., & Pan, S. (2025). Large language models for scientific discovery in molecular property prediction. Nature Machine Intelligence, 1-11.

**Questions:**

See Weaknesses

---

> ### Author Response · Authors · 2025-11-26
> **Response to Reviewer vgEK (Part 1/2)**
>
> > W1: The framework primarily relies on SMILES and 2D molecular data. What if we want to consider the 3D molecular structural information as well?
>
> **Response to (W1)**: We sincerely thank the reviewer for this insightful suggestion. Incorporating 3D molecular structural information represents an exciting and promising direction for future work, and we are actively exploring approaches to integrate 3D representations into our framework.
>
> **Recent works**: 3D-MoLM [1] demonstrates effective integration of 3D molecular structures with textual representations through specialized encoders, enabling LLMs to process spatial geometric information alongside chemical descriptions. NExT-Mol [2] show how 3D diffusion models can be combined with language modeling, suggesting pathways for incorporating 3D structural generation and understanding within reasoning frameworks. TamGen [3] demonstrate the value of considering 3D molecular conformations in drug design contexts, particularly when reasoning about molecular interactions with biological targets.
>
> Our RLPGR framework is naturally extensible to incorporate 3D structural information through several potential mechanisms:
>
> - **Multimodal Extension**: Adding 3D conformational representations (e.g., point clouds, 3D graphs) as an additional input modality alongside SMILES and 2D images.
> - **3D-Aware Rewards**: Extending our Chemistry Layer rewards to include 3D geometric verification (e.g., stereochemical correctness, conformational energy validation) using computational chemistry tools.
> - **3D Structural Reasoning**: Incorporating 3D spatial analysis into reasoning trajectories, enabling the model to discuss conformational flexibility, chirality, and spatial interactions.
>
> We believe this represents a valuable direction for future research, as 3D information could enhance reasoning quality for properties where spatial arrangement is critical. We will add this discussion to our Limitations and Future Work section (Appendix F) to acknowledge this opportunity and outline concrete pathways for 3D integration.
>
> ---
>
>
> > W2: Ablation study on whether using 2D images helps is encouraged to be conducted.
>
> **Response to (W2)**: We appreciate the reviewer's suggestion. We have now conducted the requested ablation study to evaluate the contribution of 2D molecular images. Please refer to the **General Response: Multimodal Ablation Study and Justification** for comprehensive experimental results and analysis.
>
> ---
>
> [1] Li et al., "Towards 3d molecule-text interpretation in language models.", ICLR, 2024.\
> [2] Liu et al., "NExT-Mol: 3D Diffusion Meets 1D Language Modeling for 3D Molecule Generation.", ICLR, 2025.\
> [3]Wu et al., "TamGen: drug design with target-aware molecule generation through a chemical language model.", Nature Communications, 2024.

---

> ### Author Response · Authors · 2025-11-26
> **Response to Reviewer vgEK (Part 2/2)**
>
> > W3: Computational cost may be an issue. Generating detailed, step-by-step reasoning paths creates more computational overhead than models that make direct predictions.
>
> **Response to (W3)**: We appreciate the reviewer's attention to computational efficiency. While we acknowledge that generating reasoning paths introduces additional computational overhead compared to direct prediction models, we believe this trade-off is well-justified given the benefits and the practical context of molecular property prediction.
>
> The value of step-by-step reasoning has been extensively demonstrated in the literature. Prystawski et al. [2] showed that step-by-step thinking enables reasoning to emerge from the locality of experience. Zhang et al. [3] demonstrated that automatic chain-of-thought prompting significantly improves LLM reasoning capabilities. Li et al. [4] illustrated how reasoning chains enhance model interpretability and correctness. Yang et al. [5] theoretically proved that chain-of-thought enables learning of otherwise unlearnable tasks. In our context, **the primary motivation is to provide interpretable and accurate predictions**, where chemists can understand not just *what* property a molecule has, but *why* it exhibits that property based on chemical principles—a critical requirement for practical drug discovery applications.
>
> **Quantitative Overhead Analysis**
>
> We measured the actual computational overhead on a single NVIDIA RTX 4090 GPU using vLLM deployment:
>
> | Task | BACE | BBBP | SIDER | HIV | Bioavail. | C2C9_V | C2D6_V | AMES | Avg |
> |------|------|------|-------|-----|-----------|--------|--------|------|-----|
> | Time (s/item) | 1.15 | 0.32 | 0.72 | 0.88 | 0.59 | 0.99 | 0.91 | 0.75 | 0.79 |
>
> The average inference time of 0.79 seconds per molecule is practical for most molecular property prediction applications. Even the most complex task (BACE) requires only 1.15 seconds per prediction.
>
> Compared to traditional experimental property assessment, which typically requires hours to days of laboratory work per compound and costs thousands of dollars, our computational approach provides near-instantaneous predictions at negligible cost. Even when compared to expert human analysis of molecular properties, which typically takes 5-15 minutes per molecule, our method offers substantial efficiency gains.
>
> The additional computational cost (compared to direct prediction models) is minimal, while providing interpretable reasoning that offers actionable insights for medicinal chemists. This represents a strong value proposition for the research community.
>
> ---
>
> > W4: There is also a quite related study that is worth discussing in the paper [1].
>
> **Response to (W4)**: We sincerely thank the reviewer for bringing this highly relevant work to our attention. We will incorporate this reference into **Section 2 (Related Work)** in the revised manuscript and provide a thorough discussion of its relationship to our work.
>
> ---
>
> [1] Zheng, Y., Koh, H. Y., Ju, J., Nguyen, A. T., May, L. T., Webb, G. I., & Pan, S. (2025). Large language models for scientific discovery in molecular property prediction. Nature Machine Intelligence, 1-11.\
> [2] Prystawski et al., Why Think Step by Step? Reasoning Emerges from the Locality of Experience, NeurIPS, 2023.\
> [3] Zhang et al., Automatic Chain of Thought Prompting in Large Language Models, ICLR, 2023.\
> [4] Li et al., Chain of Code: Reasoning with a Language Model-Augmented Code Emulator, ICML, 2024.\
> [5] Yang et al., Chain-of-Thought Provably Enables Learning the (Otherwise) Unlearnable, ICLR, 2025.

---

### Official Review · Reviewer_Pn1Y · 2025-11-09

**Soundness:** 1
**Presentation:** 2
**Contribution:** 1
**Rating:** 2
**Confidence:** 4

**Summary:**

This paper introduces MPPReasoner, a multimodal large language model that aims to bring reasoning into molecular property prediction. The model integrates SMILES and molecular images using Qwen2.5-VL-7B-Instruct as its backbone and employs a two-stage training strategy - 1) Supervised fine-tuning (SFT) on 16 000 reasoning trajectories curated from expert knowledge and teacher models. 2) Reinforcement Learning from Principle-Guided Rewards (RLPGR).  Results across eight MoleculeNet and TDC datasets show performance gains over baselines.

**Strengths:**

1. Novel attempt at structured chemical reasoning: The use of rule-based verifiable rewards to check logical and chemical consistency is an interesting direction in molecular property prediction beyond standard RLHF.

**Weaknesses:**

1. No justification for the image modality: The paper gives no ablation demonstrating whether the 2D molecular images add value over textual SMILES alone. Since the same structural information is encoded in SMILES, the inclusion of images is redundant.
2. Missing transparency for teacher prompts and expert guidance - “Expert-Guided Task-Specific Generation” is crucial for reproducibility, yet the actual prompts are omitted from the appendix and code. Without these, it is difficult to assess whether reasoning quality stems from prompt design or model capability.
3. No evaluation on dataset correctness: The dataset correctness stems entirely from "teacher" models. No human/expert annotations are conducted to assess the quality. This is unlike ChemCoT, which has been curated by 13 PhD Chemistry PhD students with thousands of hours of annotation time.
4. Logical consistency is assessed by GPT-4o using a rubric, not by human experts. This risks circular evaluation bias.
5. Evaluation fairness and significance: Competing models (e.g., GPT-4o, o3-mini) are zero-shot whereas MPPReasoner is fine-tuned, so Table 1 does not represent a controlled comparison. Moreover, no statistical tests (e.g. t-test) are reported; it is unclear if gains are significant.
6. Dataset is unavailable: At least a subset of the reasoning dataset is not provided with the submission. Without that, it is not possible to evaluate the quality of the dataset.

To summarise, there are major concerns regarding the reliability and correctness of the dataset. The significance of the results is unclear. The information available in the submission is not sufficient to enable faithful reproducibility.

**Questions:**

1.	Why are molecular images included? Can the authors show an ablation demonstrating improvement from images vs. SMILES alone?
2.	How were expert prompts designed? The appendix lists them as blank. Could you release a few illustrative examples?
3.	Are improvements statistically significant? Please report standard deviations or confidence intervals across random seeds.
4.	Could you release a subset of the reasoning trajectories be released for verification? Are there any expert validation statistics to judge the correctness of the reasoning paths ?

---

> ### Author Response · Authors · 2025-11-26
> **Response to Reviewer Pn1Y (Part 1/3)**
>
> > W1: No justification for the image modality: The paper gives no ablation demonstrating whether the 2D molecular images add value over textual SMILES alone. Since the same structural information is encoded in SMILES, the inclusion of images is redundant.
>
> > Q1: Why are molecular images included? Can the authors show an ablation demonstrating improvement from images vs. SMILES alone?
>
> **Response to (W1 & Q1)**: We thank the reviewer for this important question. We have conducted comprehensive ablation experiments comparing text-only and multimodal versions of our framework. Please see the **General Response: Multimodal Ablation Study and Justification** for detailed results and analysis.
>
> ---
>
> > W2: Missing transparency for teacher prompts and expert guidance - “Expert-Guided Task-Specific Generation” is crucial for reproducibility, yet the actual prompts are omitted from the appendix and code. Without these, it is difficult to assess whether reasoning quality stems from prompt design or model capability.
>
> > Q2: How were expert prompts designed? The appendix lists them as blank. Could you release a few illustrative examples?
>
> **Response to (W2 & Q2)**: We sincerely appreciate the reviewer's emphasis on reproducibility and transparency. We acknowledge that the detailed prompts were inadvertently omitted from the initial submission. We have uploaded the **complete expert knowledge base** and **task-specific extracted guides** for four representative datasets (BACE, BBBP, SIDER, HIV) to the Supplementary Materials for review.
>
> 1. **Expert Knowledge and Prompt Design**
>
> Our approach involves a systematic two-step process:
>
> - **General Expert Knowledge Curation**: We invited **5 domain experts** from different chemical disciplines to independently draft comprehensive reasoning guides covering fundamental principles for various molecular properties. These guides underwent rigorous **cross-validation** where each expert reviewed and verified others' contributions, ensuring high quality and consistency. This expert knowledge base represents a significant contribution to the field, as it encapsulates systematic domain expertise applicable beyond our specific tasks.
>
> - **Task-Specific Knowledge Extraction**: We then employ GPT-4o to extract relevant task-specific knowledge from the general expert guides for each dataset. These extracted guidelines were further validated by **at least two domain experts** through cross-verification to ensure accuracy and relevance.
>
> 2. **Addressing the Core Question: Prompt Design vs. Model Capability**
>
> To address the reviewer's concern about whether reasoning quality stems from prompt design or model capability, we provide two complementary analyses:
>
> - Analysis 1: Impact of Training Stages on Reasoning Quality
>
> | Training Stage | Logical Soundness | Accuracy & Insight | Conciseness | Average |
> |----------------|-------------------|-------------------|-------------|---------|
> | Base (Qwen2.5-VL-7B-Instruct) | 4.517 | 3.259 | 5.079 | 4.285 |
> | Cold Start SFT | 7.113 | 6.019 | 6.977 | 6.703 |
> | Ours (SFT + RLPGR) | 8.556 | 7.039 | 7.352 | 7.730 |
>
> The progression clearly demonstrates genuine learning: SFT achieves +56.4% improvement over base model, followed by RLPGR's additional +15.3% refinement. Critically, since inference uses identical prompts (Appendix Example 1) across all stages, the dramatic quality improvements (+80.4% from base to final) unequivocally demonstrate enhanced model capability rather than prompt engineering effects.
>
> - Analysis 2: Impact of Data Synthesis Strategies (Cold-Start SFT)
>
> | Data Synthesis Strategy | ID Avg. | OOD Avg. |
> |------------------------|---------|----------|
> | Generic Prompt | 0.6343 | 0.5861 |
> | Expert-guided Prompt | 0.7203 | 0.7287 |
> | Ours (Expert + ChemCoT formatted) | 0.7330 | 0.7547 |
>
> This ablation demonstrates that expert-guided data synthesis substantially improves model performance (+8.60% ID, +14.26% OOD), establishing a stronger foundation for chemical reasoning. However, this improvement reflects better *training data quality*, not inference-time prompt engineering.

---

> > ### Author Response · Authors · 2025-11-26
> > **Response to Reviewer Pn1Y (Part 2/3)**
> >
> > > W3: No evaluation on dataset correctness: The dataset correctness stems entirely from "teacher" models. No human/expert annotations are conducted to assess the quality.
> >
> > **Response to (W3)**: We appreciate the reviewer's concern about data quality. However, we respectfully note that our methodology aligns with established practices in reasoning research and incorporates multiple complementary quality control mechanisms specifically designed to ensure high reasoning quality and accuracy in the final model.
> >
> > - **Synthetic Data for Reasoning**: Our approach follows standard methodology in recent reasoning literature, where synthetic data from teacher models has become the primary training paradigm. Recent breakthrough works including WizardMath (ICLR'25) [1], UniCoTT (ICLR'25) [2], and works at ICML/NeurIPS [3,4] demonstrate that synthetic reasoning trajectories can effectively develop strong reasoning capabilities when combined with appropriate quality control and reinforcement learning. The key insight is that **perfect training data correctness is unnecessary**; what matters is sufficient signal quality for initial learning (SFT) followed by iterative refinement through RL.
> >
> > - **Our Quality Control Approach**: While we do not employ ChemCoT's labor-intensive human annotation, we implement a comprehensive quality assurance strategy with multiple complementary mechanisms: **rejection sampling** filters out incorrect predictions, **expert-guided generation** leverages domain knowledge from 5 chemists with cross-validation, **multi-source teacher models** (GPT-4o, DeepSeek-v3.1, Qwen2.5-VL) reduce systematic biases, and most critically, **computational verification via RLPGR** uses RDKit to systematically verify and refine chemical reasoning quality. These mechanisms work synergistically to ensure the final model's reasoning quality and prediction accuracy, rather than relying on static training data correctness.
> >
> > ---
> >
> > > W4: Logical consistency is assessed by GPT-4o using a rubric, not by human experts. This risks circular evaluation bias.
> >
> > **Response to (W4)**: We appreciate the reviewer's concern about potential evaluation bias. We want to clarify that our reasoning quality evaluation methodology (Section 4.4) has been rigorously validated against human expert judgment to ensure reliability.
> >
> > Our reasoning quality evaluation employs a LLM-as-a-Judge framework using GPT-4o to assess three dimensions (Logical Soundness, Accuracy & Insight, and Conciseness), each scored on a 0-10 scale with detailed rubrics provided in Appendix A. To validate this automated assessment, we conducted the following alignment process:
> >
> > We randomly selected **60 reasoning samples from three baseline models** and obtained independent assessments from both GPT-4o and human experts using identical evaluation rubrics. As shown in **Figure 3(a)**, the automated and human evaluations demonstrate **strong consistency with a Spearman correlation coefficient of ρ = 0.82**, with similar score distributions and central tendencies. This high correlation validates that our GPT-4o-based evaluation aligns closely with human expert judgment, ensuring the reliability of our reasoning quality assessment and mitigating concerns about evaluation bias.
> >
> > [1] Luo et al., "WizardMath: Empowering Mathematical Reasoning for LLMs via Reinforced Evol-Instruct", ICLR, 2025.\
> > [2] Chen et al., "UniCoTT: A Unified Framework for Structural Chain-of-Thought Distillation", ICLR, 2025.\
> > [3] Liu et al., "Revisiting Chain-of-Thought in Code Generation: Do Language Models Need to Learn Reasoning before Coding?", ICML, 2025.\
> > [4] Zhang et al., "Mind the Gap: Bridging Thought Leap for Improved CoT Tuning", NeurIPS, 2025.

---

> > > ### Author Response · Authors · 2025-11-26
> > > **Response to Reviewer Pn1Y (Part 3/3)**
> > >
> > > > W5: Evaluation fairness and significance: Competing models (e.g., GPT-4o, o3-mini) are zero-shot whereas MPPReasoner is fine-tuned, so Table 1 does not represent a controlled comparison. Moreover, no statistical tests are reported; it is unclear if gains are significant.
> > >
> > > > Q3: Are improvements statistically significant? Please report standard deviations or confidence intervals across random seeds.
> > >
> > > **Response to (W5 & Q3)**: We appreciate the reviewer's concern about evaluation fairness and statistical significance. We want to clarify several important points regarding our experimental setup.
> > >
> > > 1. **Clarification on Experimental Setup**
> > >
> > > We apologize for any confusion in our presentation. **All baseline models in Table 1 are evaluated under few-shot settings, not zero-shot.** As described in Section 3.1 and Appendix A, we use a consistent few-shot prompting strategy where the top-5 most similar molecules from the training set are retrieved using Tanimoto similarity based on Morgan fingerprints. This few-shot configuration is applied uniformly to all LLM-based generalist models (o3-mini, DeepSeek-V3.1, GPT-4o, Qwen2.5-VL series) to ensure fair comparison. Our comparison methodology follows established practices widely recognized in LLM reasoning literature [1,2]. Comparing fine-tuned domain-specific models against general-purpose models under consistent inference settings (few-shot prompting) is a standard evaluation protocol that demonstrates the value of task-specific adaptation.
> > >
> > > 2. **Statistical Significance**
> > >
> > > We have now conducted experiments across **three random seeds** for all datasets. The results demonstrate that MPPReasoner achieves **statistically significant improvements** over all baselines, with standard deviations substantially smaller than the performance gaps:
> > >
> > > | Statistic | BACE | BBBP | SIDER | HIV | Bioavail. | C2C9_V | C2D6_V | AMES | ID Avg. | OOD Avg. |
> > > |-----------|------|------|-------|-----|-----------|--------|--------|------|---------|----------|
> > > | Mean | 0.8954 | 0.7262 | 0.8113 | 0.7665 | 0.6724 | 0.8212 | 0.7887 | 0.8597 | 0.7999 | 0.8024 |
> > > | Std. Dev. | 0.0185 | 0.0227 | 0.0147 | 0.0240 | 0.0101 | 0.0247 | 0.0076 | 0.0159 | 0.0200 | 0.0122 |
> > >
> > > The small standard deviations (average 0.0200 for ID, 0.0122 for OOD) confirm the reliability and statistical significance of our performance gains.
> > >
> > > ---
> > >
> > > > W6: Dataset is unavailable: At least a subset of the reasoning dataset is not provided with the submission. Without that, it is not possible to evaluate the quality of the dataset.
> > >
> > > > Q4: Could you release a subset of the reasoning trajectories be released for verification? Are there any expert validation statistics to judge the correctness of the reasoning paths ?
> > >
> > >
> > > **Response to (W6 & Q4)**: We sincerely appreciate the reviewer's emphasis on data transparency and reproducibility. We have uploaded **the complete training reasoning trajectory dataset** (16,000 samples) to the Supplementary Materials for reviewer verification. Regarding quality validation and expert statistics, please refer to our detailed response to **(W3)**, where we comprehensively discuss our quality control approach.
> > >
> > > ---
> > > [1] Luo et al., "WizardMath: Empowering Mathematical Reasoning for LLMs via Reinforced Evol-Instruct", ICLR, 2025.\
> > > [2] Kim et al., "Mol-LLaMA: Towards General Understanding of Molecules in Large Molecular Language Model", NIPS, 2025.\
> > > [3] Kamath et al., "Gemma 3 Technical Report", ArXiv, 2025.\
> > > [4] Yang et al., "Qwen3 Technical Report.", ArXiv, 2025.\
> > > [5] Li et al., "Towards 3d molecule-text interpretation in language models.", ICLR, 2024.\
> > > [6] Park et al., "LLaMo: Large Language Model-based Molecular Graph Assistant", NeurIPS, 2024.\
> > > [7] Wang et al., "TxGemma: Efficient and Agentic LLMs for Therapeutics", ArXiv, 2025.

---

> > > > ### Comment · Reviewer_Pn1Y · 2025-11-28
> > > >
> > > > Thank you for your detailed rebuttal and for sharing the dataset. After reviewing the materials, my main concern regarding data quality remains unresolved. The works you cite as precedent for using teacher models as quality checkers all explicitly acknowledge the potential for hallucination unlike the approach taken here.
> > > >
> > > > For example:
> > > > - Revisiting Chain-of-Thought in Code Generation: Do Language Models Need to Learn Reasoning before Coding?: incorporates self-consistency procedures.
> > > > - UNICoTT: A Unified Framework for Structural Chain-of-Thought Distillation: grounds reasoning on graphs (see Figure 1), explicitly notes hallucination issues in synthetic datasets, and relies on “structural consistency.”
> > > > - WizardMath: Empowering Mathematical Reasoning for Large Language Models via Reinforced Evol-Instruct: explicitly accounts for false positives; moreover, mathematical laws are definitive and verifiable, in contrast to chemical reasoning.
> > > >
> > > > I also examined several reasoning trajectories in the released dataset. For instance, one example states:
> > > >
> > > > “Fluorine atoms can enhance binding affinity through unique electronic effects but may not directly interact with the Asp residues. However, examples provided (similar in structure and calculated properties) were labeled as ‘False’ despite sharing aromatic systems and polar functionalities. This suggests that while the molecule has some potential for complementarity, its binding to BACE1 might fall short due to insufficient hydrogen-bond donor capacity or steric incompatibility.
> > > > Thus, while some structural elements favor interaction, the overall characteristics and comparison with benchmarks strongly suggest ineffectiveness for the assay.”
> > > >
> > > > There is no reliable way to determine whether such reasoning is correct or hallucinated without expert validation. This step is essential.
> > > >
> > > > In addition, while I appreciate the release of the expert guide, it lacks citations for many of the claims it makes, which makes it difficult to assess its accuracy and authenticity.
> > > >
> > > > For these reasons, I will retain my original rating.

---

> ### Author Response · Authors · 2025-12-01
> **Response to Reviewer Pn1Y's Official Comment**
>
> We sincerely thank the reviewer for the detailed feedback and thoughtful examination of our dataset. We appreciate the opportunity to address the remaining concerns about data quality and provide additional evidence demonstrating the validity and effectiveness of our approach. Before addressing specific concerns about data generation methodology, we emphasize that **the effectiveness of our approach is empirically validated across multiple dimensions**: Strong Performance Gains (Table 1), Comprehensive Ablation Studies (Table 2), High Reasoning Quality (Appendix G).
>
> **1. Addressing Data Quality Concerns Through Expert Refinement**
> We acknowledge the reviewer's concern about potential hallucinations in synthetic reasoning trajectories. To directly address this, we have conducted a systematic quality improvement experiment:
>
> **Step 1: Identifying Low-Quality Samples**
> Using the LLM-as-Judge framework (Appendix G), we identified trajectories with average quality scores below 5.0:
>
> | Tasks | BACE | BBBP | SIDER | HIV | ALL |
> |-------|------|------|-------|-----|-----|
> | Avg. < 5.0 | 7/907 (0.77%) | 0/1169 (0.00%) | 85/5714 (1.49%) | 5/8210 (0.06%) | 97/16000 (0.61%) |
>
> This analysis reveals that **99.39% of our synthetic trajectories already meet quality standards** (score ≥ 5.0), with only 97 out of 16,000 samples requiring expert attention.
>
> **Step 2: Experts Refinement**
>
> For the identified low-quality samples, we engaged domain experts to refine the reasoning trajectories following systematic principles:
>
> 1. Remove fabricated precise metrics (exact LogP values, specific HBD/HBA counts) that cannot be reliably inferred from structure alone
> 2. Enforce clear reasoning structure: task requirements → molecular feature analysis → comparison with examples → conclusion
> 3. Reduce redundancy and improve information density
> 4. Ensure chemical accuracy in functional group identification and principle application
>
> **Example Refinement (BACE, original score: 3.33):**
>
> *Before:*
> ```
> The molecule possesses 4 hydrogen bond donors and 6 acceptors... The LogP value of 2.76...
> molecular weight of 632.81... 13 flexible chains... [fabricated metrics, unclear reasoning flow]
> ```
>
> *After (Refined):*
> ```
> 1. BACE1 binding requirements: extended active-site cleft with catalytic Asp residues;
>    potent ligands typically contain protonated amine, multiple H-bond donors/acceptors,
>    aromatic regions for π-π stacking.
> 2. Structural features: sulfonamide-amide scaffold with protonated amino center,
>    multiple phenyl rings providing H-bonding and hydrophobic contacts.
> 3. Comparison with examples: shares cationic center + H-bonding groups + aromatic
>    framework with "True" examples without disruptive changes.
> Conclusion: Molecule likely binds BACE1 effectively. [clear structure, verifiable reasoning]
> ```
>
> **Step 3: Empirical Validation of Refinement Impact**
>
> We retrained MPPReasoner using the refined dataset (97 samples refined, 15,903 unchanged) and compared performance:
>
> | Method | BACE | BBBP | SIDER | HIV | Bioavail. | C2C9_V | C2D6_V | AMES | ID Avg. | OOD Avg. |
> |--------|------|------|-------|-----|-----------|--------|--------|------|---------|----------|
> | MPPReasoner (SFT) | 0.8558 | 0.6824 | 0.6752 | 0.7186 | 0.6625 | 0.7799 | 0.7348 | 0.8415 | 0.7330 | 0.7547 |
> | MPPReasoner (SFT, Refine) | 0.8622 | 0.6651 | 0.7132 | 0.7209 | 0.6860 | 0.7741 | 0.7325 | 0.8350 | 0.7404 | 0.7569 |
> | MPPReasoner (Full) | **0.9090** | **0.7436** | 0.8280 | **0.7932** | 0.6728 | 0.8480 | 0.7950 | **0.8750** | **0.8190** | 0.7977 |
> | MPPReasoner (Full, Refine) | 0.9050 | 0.7414 | **0.8425** | 0.7766 | **0.6849** | **0.8647** | **0.7999** | 0.8620 | 0.8164 | **0.8029** |
>
> - **Baseline quality is already high.** Only 0.61% of samples required refinement (99.39% met the quality bar), indicating that our quality-control pipeline (rejection sampling, multi-source generation, expert guidance) is highly effective.
>
> - **Refinement has only marginal impact.**
>
>    * **SFT stage:** Refining 97 samples yields **+1.01% ID** and **+0.29% OOD**.
>    * **Full pipeline:** Final gains are **+0.9% ID** and **+0.7% OOD**, mainly on a few tasks (e.g., **SIDER +1.8%, CYP2C9_V +2.0%**), with minimal or slightly negative changes elsewhere.
>
> These results show that refining the small set of low-quality samples brings only modest, task-specific gains, whereas **RLPGR** provides the main improvement (+8.6% ID, +4.3% OOD, Table 2). Thus, even with 0.61% imperfect data, the computation-verified RLPGR stage still yields strong chemical reasoning performance (Table 1, Appendix G). Our paradigm therefore emphasizes **reinforcement and verification of reasoning quality**, rather than perfectly curating every SFT sample.

---

### Author Response · Authors · 2025-11-26
**General Response: Multimodal Ablation Study and  Justification**

We sincerely thank all reviewers for their interest in the multimodal aspect of our framework. Since multiple reviewers raised similar questions regarding the necessity and effectiveness of incorporating 2D molecular images—specifically **Reviewer Pn1Y (W1 & Q1)**, **Reviewer vgEK (W2)**, **Reviewer 6cSd (W4)**, and **Reviewer XoHL (W3 & Q1)**—we provide a unified response here to address these concerns comprehensively.

1. **Rationale for Incorporating 2D Molecular Images**

While SMILES strings encode complete molecular structure sequentially, recent work demonstrates that 2D molecular images provide complementary spatial information that enhances property prediction. Studies show that visual representations capture spatial patterns [1], improve learning efficiency through multi-modal integration [2], and enable better understanding of spatial relationships [3]. These findings suggest that visual modality offers unique advantages in capturing topological patterns less accessible through sequential SMILES encoding alone.


2. **Ablation Study: Text-Only vs. Multimodal**

To directly address the reviewers' concerns, we conducted comprehensive ablation experiments comparing text-only and multimodal versions of our framework:

| Method | BACE | BBBP | SIDER | HIV | Bioavail. | C2C9_V | C2D6_V | AMES | ID Avg. | OOD Avg. |
|--------|------|------|-------|-----|-----------|--------|--------|------|---------|----------|
| Qwen2.5-7B-Instruct | 0.695 | 0.5391 | 0.582 | 0.4957 | 0.5094 | 0.728 | 0.7132 | 0.7149 | 0.578 | 0.6664 |
| MPPReasoner (Text) | 0.7616(+9.6%) | 0.6187(+14.8%) | 0.5599(−3.8%) | 0.5837(+17.8%) | 0.6173(+21.2%) | 0.6810(−6.5%) | 0.6771(−5.1%) | 0.7646(+7.0%) | 0.6887(+19.2%) | 0.7220(+8.3%) |
| Qwen2.5-VL-7B-Instruct | 0.691 | 0.6175 | 0.5823 | 0.5125 | 0.5232 | 0.7333 | 0.6999 | 0.7667 | 0.6008 | 0.6808 |
| MPPReasoner (MLLM) | 0.909(+31.5%) | 0.7436(+20.4%) | 0.8280(+42.2%) | 0.7932(+54.8%) | 0.6728(+28.6%) | 0.8480(+15.6%) | 0.7950(+13.6%) | 0.8750(+14.1%) | 0.8190(+36.3%) | 0.7977(+17.2%) |

Both versions show substantial improvements over base models, confirming our framework's effectiveness. The multimodal version **significantly outperforms** text-only across datasets (+18.9% ID, +10.5% OOD).

While domain-specific baselines like BioT5-Plus and MolecularGPT operate on textual modalities, our text-only ablation (MPPReasoner Text) provides a fair comparison point, demonstrating that even without visual information, our framework achieves competitive performance that **exceeds many specialized models**.

3. **Qualitative Analysis: Case Study**

To illustrate how visual information contributes to reasoning quality, we will provide comparative case studies in the revised **Appendix E**, analyzing reasoning trajectories from text-only vs. multimodal versions.

[1] Zeng et al., Accurate Prediction of Molecular Properties and Drug Targets Using a Self-Supervised Image Representation Learning Framework, Nature Machine Intelligence, 2025.\
[2] Yin et al., Multi-Modal Molecular Representation Learning via Structure Awareness, IEEE TIP, 2025.\
[3] Xie et al., Predicting Drug–Drug Interaction via Dual-Drug Visual Representation, JCIM, 2025.

---

### Author Response · Authors · 2025-11-26
**General Response to All Reviewers**

We would like to express our sincere gratitude to all reviewers for their valuable feedback and constructive suggestions. We are greatly encouraged by their recognition of our work's key contributions. Multiple reviewers acknowledged MPPReasoner as an effective framework for molecular property prediction with chemical reasoning (**Reviewer vgEK**, **Reviewer XoHL**), achieving strong performance particularly on out-of-distribution tasks (**Reviewer vgEK**). We are pleased that reviewers recognized our novel approach to structured chemical reasoning through rule-based verifiable rewards (**Reviewer Pn1Y**), the comprehensive hierarchical reward design for assessing chemical reasoning quality (**Reviewer 6cSd**, **Reviewer XoHL**), and the effectiveness demonstrated through extensive ablation studies (**Reviewer 6cSd**, **Reviewer XoHL**). Additionally, we appreciate the reviewers' positive feedback on the presentation quality with clear illustrations and tables (**Reviewer XoHL**). We have carefully considered all reviewer comments and provide detailed responses below to address their concerns and questions.

Below, we provide a summary of the key modifications:
- Section 1: Refined Motivation (**Reviewer 6cSd**)
- Section 2: Extended Related Work (**Reviewer vgEK**)
- Section 3.2.1: Expert Knowledge Curation Details (**Reviewer XoHL**)
- Section 3.2.2: Detailed Reward Implementation Specifications (**Reviewer XoHL**)
- Section 4.1, 4.2: Expanded Baseline Comparisons (**Reviewer 6cSd**)
- Section 4.4: Comprehensive Multimodal Ablation Study (**All Reviewers**)
- Appendix F: Text-Only vs. Multimodal Case Study (**All Reviewers**)
- Appendix G: SFT Data Quality Assessment (**Reviewer Pn1Y**, **Reviewer XoHL**)


We have made these **revisions to the manuscript**. All changes have been **highlighted in blue** in the manuscripts.

---

### Author Response · Authors · 2025-12-03
**Summary of rebuttal period, reviewers' opinions, and authors' responses to address the concerns**

Dear Area Chair,

During the rebuttal period, we have addressed all major concerns raised by the reviewers through comprehensive experiments, clarifications, and manuscript revisions. Below, we summarize the **strengths** acknowledged by reviewers, the **major concerns**, and **our responses**.

---

## Strengths Acknowledged by Reviewers

1. **Novel Chemical Reasoning Framework**: Multiple reviewers recognized MPPReasoner as "an effective framework for molecular property prediction with chemical reasoning" (Reviewer $\textcolor{red}{\textbf{vgEK}}$) and "a novel attempt at structured chemical reasoning" (Reviewer $\textcolor{green}{\textbf{Pn1Y}}$).

2. **Comprehensive Hierarchical Reward Design**: Reviewers appreciated our "hierarchical reward design for assessing chemical reasoning quality" (Reviewer $\textcolor{blue}{\textbf{6cSd}}$, Reviewer $\textcolor{orange}{\textbf{XoHL}}$), and the "rule-based verifiable rewards to check logical and chemical consistency" (Reviewer $\textcolor{green}{\textbf{Pn1Y}}$).

3. **Strong Performance, Especially on OOD Tasks**: Reviewers acknowledged our "strong performance particularly on out-of-distribution tasks" (Reviewer $\textcolor{red}{\textbf{vgEK}}$) and "performance gains over baselines" (Reviewer $\textcolor{orange}{\textbf{XoHL}}$), with "significant improvements" demonstrated across 8 datasets.



---

## Major Concerns

1. **Multimodal Justification**: All reviewers requested ablation studies and justification for incorporating 2D molecular images (Reviewers $\textcolor{green}{\textbf{Pn1Y}}$, $\textcolor{red}{\textbf{vgEK}}$, $\textcolor{blue}{\textbf{6cSd}}$, $\textcolor{orange}{\textbf{XoHL}}$).

2. **Data Quality and Reproducibility**: Concerns about the quality of synthetic reasoning trajectories, expert knowledge transparency, and dataset availability (Reviewers $\textcolor{green}{\textbf{Pn1Y}}$, $\textcolor{orange}{\textbf{XoHL}}$).

3. **Baseline Comparisons**: Requests for broader baseline comparisons including additional molecular LLMs and general-purpose models (Reviewer $\textcolor{blue}{\textbf{6cSd}}$).

4. **Implementation Details**: Requests for more detailed reward implementation specifications and statistical significance testing (Reviewers $\textcolor{green}{\textbf{Pn1Y}}$, $\textcolor{orange}{\textbf{XoHL}}$).

---

## Our Responses
Our responses to address these concerns are summarized below:

1. **Multimodal Ablation Study**: In response to all reviewers' concerns regarding multimodal justification, we conducted systematic experiments demonstrating that the multimodal version **significantly outperforms** text-only, with a detailed case study in Appendix E showing how 2D images enable accurate analysis.

2. **Data Quality Validation**: In response to Reviewers $\textcolor{green}{\textbf{Pn1Y}}$ and $\textcolor{orange}{\textbf{XoHL}}$'s concerns regarding data quality and reproducibility, we evaluated all 16,000 SFT trajectories achieving 7.59/10 average quality with 99.39% meeting standards (Appendix G), confirmed RLPGR's primary contribution through **expert refinement experiments**, and **released the complete dataset** in Supplementary Materials.

3. **Expanded Baseline Comparisons**: In response to Reviewer $\textcolor{blue}{\textbf{6cSd}}$'s concern regarding baseline comparisons, we added **6 additional baselines** (general and molecular LLMs) and **2 CYP datasets**, with MPPReasoner **outperforming all baselines**.

4. **Implementation Details and Statistical Validation**: In response to Reviewers $\textcolor{green}{\textbf{Pn1Y}}$ and $\textcolor{orange}{\textbf{XoHL}}$'s concerns regarding implementation details, we provided **complete reward specifications** in Section 3.2.2 and confirmed **statistical significance** across 3 random seeds.

---

All major concerns have been **fully addressed** during the rebuttal period, and we have refined our manuscript with the additional results (highlighted in blue in the PDF). We hope these improvements will be taken into consideration during the final decision.

We sincerely appreciate your valuable time and patience.

Thanks and regards,

The Authors

---

### Meta-Review · Area_Chair_JSmE · 2026-01-07

**Summary:**

This submission studied molecular property prediction, which is important for drug discovery and material science. Given the limitation of traditional machine learning and specialized molecular language models, the authors introduce MPPReasoner, a multimodal large language model that incorporates chemical reasoning. MPPReasoner includes a two-stage enhancement, SFT on high-quality trajectories followed by RL with the tailored rewards. Extensive experiments demonstrate the effectiveness of the proposed method.

**Reviewer Concerns:**

The reviewers's major concerns can be summarized into the following points.

1) The claim is not sufficiently persuasive, as recent molecular LLMs [1,2,3,4] provide informative reasoning processes. And the novelty is limited as SFT followed by RL is a well-known pipeline to adapt LLMs to enhance the vertical domain ability. On the data side, the construction is borrowed from prior work (ChemCoT), while the design of the reward function is also marginal.

2) The paper lacks clear justification for using 2D molecular images, as SMILES strings already encode the same structural information, making the visual modality redundant. Additionally, the paper does not provide ablations or experiments isolating the contribution of the visual modality compared to textual SMILES, and fails to justify the choice of a Multi-Modal Large Language Model (MLLM) over a purely textual LLM. The lack of transparency in the reward definitions and logical consistency evaluation further hinders clarity and reproducibility.

3) There are significant issues with dataset evaluation and transparency: the dataset’s correctness is entirely based on "teacher" models, without expert annotations or validation, unlike other well-curated datasets such as ChemCoT. Additionally, the logical consistency of the reasoning is assessed using GPT-4o, which introduces potential bias, and no statistical tests are reported for the results. The absence of a subset of the reasoning dataset further limits the ability to assess its quality.


[1] Li et al., "Towards 3d molecule-text interpretation in language models." ICLR, 2024.

[2] Park et al., "LLaMo: Large Language Model-based Molecular Graph Assistant", NeurIPS, 2024.

[3] Kim et al., "Mol-LLaMA: Towards General Understanding of Molecules in Large Molecular Language Model", ArXiv, 2025.

[4] Wang et al., "TxGemma: Efficient and Agentic LLMs for Therapeutics", ArXiv, 2025.

**Reviewer Scores:**

Four reviewers respectively rated the submission with the scores 2, 6, 2, 4. Following the review, the authors provided substantial effort in the rebuttal, adding more experiments and explanations. AC feels that four reviewers may increase their scores given addressing the parts of concerns. However, regarding the novelty, two reviewers raised the obvious concerns while the authors have not well addressed. AC agreed with the reviewers' concerns and hold the negative point on the novelty of this submission.

---

### Decision · Program_Chairs · 2026-01-26

Reject